

# Dimensional analysis of a landscape evolution model with incision threshold

Nikos Theodoratos[1] and James W. Kirchner[1,2]

[1]Dept. of Environmental Systems Science, ETH Zurich, Zurich, 8092, Switzerland
[2]Swiss Federal Research Institute WSL, Birmensdorf, 8903, Switzerland

*Correspondence to*: Nikos Theodoratos (theodoratos@usys.ethz.ch)

**Abstract.** The ability of erosional processes to incise into a topographic surface can be limited by a threshold. Incision thresholds affect the topography of landscapes and their scaling properties, and can introduce non-linear relations between climate and erosion with notable implications for long-term landscape evolution. Despite their potential importance, incision
thresholds are often omitted from the incision terms of landscape evolution models (LEMs) to simplify analyses. Here, we present theoretical and numerical results from a dimensional analysis of an LEM that includes terms for threshold-limited stream-power incision, linear diffusion, and uplift. The LEM is parameterized by four parameters (incision coefficient and incision threshold, diffusion coefficient, and uplift rate). The LEM's governing equation can be greatly simplified by recasting it in a dimensionless form that depends on only one dimensionless parameter, the incision-threshold number $N_\theta$.
This dimensionless parameter is defined in terms of the incision threshold, the incision coefficient, and the uplift rate, and it quantifies the reduction in the rate of incision due to the incision threshold relative to the uplift rate. Being the only parameter in the dimensionless governing equation, $N_\theta$ is the only parameter controlling the evolution of landscapes in this LEM. Thus, landscapes with the same $N_\theta$ will evolve geometrically similarly, provided that their boundary and initial conditions are normalized according to appropriate scaling relationships, as we demonstrate using a numerical experiment.
In contrast, landscapes with different $N_\theta$ values will be influenced to different degrees by their incision thresholds. Using results from a second set of numerical simulations, each with a different incision-threshold number, we qualitatively illustrate how the value of $N_\theta$ influences the topography, and we show that relief scales with the quantity $N_\theta + 1$ (except where the incision threshold reduces the rate of incision to zero).

## 1 Introduction

In the uppermost parts of drainage basins, water is not flowing over the ground surface or is flowing too weakly to incise into it. At least two kinds of limits must typically be overcome for erosion by flowing water to begin. First, sufficient drainage area must be accumulated for overland flow to start; second, this flow must exert sufficient shear stress on the surface to overcome the mechanical resistance of rocks or soils and thus mobilize sediment (e.g., Perron, 2017).

Channel-incision terms in landscape evolution models (LEMs) often capture both of these limits by including an incision threshold below which no incision occurs. For instance, if $\tau$ is the shear stress that water exerts on the bed and $\tau_\theta$ is a critical value of shear stress (equivalently, $\tau$ and $\tau_\theta$ could refer to stream power), then the rate of incision is zero for $\tau \leq \tau_\theta$ and it can be described by a term of the form $k(\tau - \tau_\theta)^\alpha$, for $\tau > \tau_\theta$, where $k$ and $\alpha$ are constants (e.g., Howard, 1994). Including such incision terms in LEMs changes the topographic properties of the landscapes that are synthesized, for example, it leads



to decreased drainage densities, more convex hillslopes, and steeper slopes (e.g., Howard, 1994; Tucker and Bras, 1998; Perron et al., 2008).

In addition, incision thresholds can have notable consequences on the relationship between climate and long-term incision rates as described, for example, by Snyder et al. (2003), Tucker (2004), Lague et al. (2005), Perron (2017), and Deal et al. (2018). Specifically, incision thresholds stop smaller events from eroding the surface. In many wet climates, the total annual streamflow is high, but small, frequent events tend to contribute most of this total; in contrast, in many dry climates, a larger fraction of the total annual streamflow tends to be contributed by rare, but intense, events (e.g., Rossi et al., 2016). Therefore, a sufficiently high incision threshold could render ineffective a larger fraction of the total precipitation in wetter climates than in drier climates. This behavior can lead to a non-linear dependence of long-term erosion rates on average precipitation; it can even lead to the counter-intuitive observation that, in some cases, larger average precipitation corresponds to smaller long-term erosion rates (e.g., DiBiase and Whipple, 2011).

Furthermore, incision thresholds can play a role in setting the smallest scales of valley dissection, which are among the fundamental scales that characterize landscapes. For instance, Horton (1945) suggested that valley dissection stops because further dissection would lead to hillslopes that are too short to yield flow that can erode the surface. Montgomery and Dietrich (1992) found that thresholds of the topographic quantity $A(|\nabla z|)^2$, where $A$ is drainage area and $|\nabla z|$ is slope, could define locations of both channel and valley heads, the former being associated with an incision threshold and the latter with the smallest scale of dissection. Perron et al. (2008) studied the spacing of valleys, a scaling property closely related to the smallest scale of dissection. They found that valley spacing is most strongly controlled by the competition between advective and diffusive processes, such as stream incision and soil creep, respectively. However, they found that incision thresholds also control valley spacing by modulating the competition between advection and diffusion.

In Theodoratos et al. (2018), we performed a scaling analysis of an incision–diffusion LEM that did not include an incision threshold. In the present study, we add an incision threshold to that LEM and examine how our analysis needs to be modified to account for this threshold. More specifically, in Theodoratos et al. (2018), we dimensionally analyzed an LEM that includes three parameters – an incision coefficient, a diffusion coefficient, and an uplift rate. For that analysis, we used three characteristic scales (of length, height, and time) that are defined in terms of the three parameters of the LEM. As we explained in detail in Theodoratos et al. (2018), because the characteristic scales depend on the model parameters and because there are three parameters and three characteristic scales, the LEM can be greatly simplified by being re-cast in a dimensionless form that has no parameters.

Adding an incision threshold to the LEM that we analyzed in Theodoratos et al. (2018) increases the number of its parameters to four (see Eq. 1 below). This leads to the question of whether the LEM with incision threshold can be dimensionally analyzed using the same three characteristic scales that we used to dimensionally analyze the LEM without incision threshold (Theodoratos et al., 2018). Here, we hypothesize that these three scales are reasonable choices even after adding an incision threshold to the LEM, and we test this hypothesis by applying these scales and examining the resulting re-scaled equations.



## 2 Dimensional analysis of LEM that includes incision threshold

### 2.1 Governing equation

We study an LEM described by the governing equation (e.g., Howard, 1994; Dietrich et al., 2003):

$$\frac{\partial z}{\partial t} = \begin{cases} D\nabla^2 z + U, & \sqrt{A}|\nabla z| \leq \theta \\ -K\left(\sqrt{A}|\nabla z| - \theta\right) + D\nabla^2 z + U, & \sqrt{A}|\nabla z| > \theta \end{cases} \quad , \tag{1}$$

where $z$ is elevation at a point with coordinates $(x, y)$ and $t$ is time; $A$, $|\nabla z|$, and $\nabla^2 z$ are topographic properties of the point,

specifically, drainage area, slope, and Laplacian curvature, respectively; $K$ is the incision coefficient, $D$ is the diffusion coefficient, $U$ is the uplift rate, and $\theta$ is the incision threshold, a threshold value of the quantity $\sqrt{A}|\nabla z|$. The four parameters $K$, $D$, $U$, and $\theta$ are all assumed to be constant in time and uniform across space. The dimensions of these variables, topographic properties, and parameters are discussed in the following subsection.

The stream-power incision term $K\left(\sqrt{A}|\nabla z| - \theta\right)$ describes the rate of incision by flowing water. It is a special case of the more general incision term $K(A^m(|\nabla z|)^n - \theta)$, where $m$ and $n$, the exponents of drainage area and slope, have values that depend on the rate law that is assumed to describe incision (such as shear stress or stream power; e.g., Dietrich et al., 2003). Here, we examine the simplified case of $m = 0.5$ and $n = 1$ because it leads to results described by much simpler formulas; however, these results are valid for generic exponents $m$ and $n$ as well, but with more complicated formulas (see Appendix A

of Theodoratos et al. (2018) for results pertaining to an LEM without incision threshold and with generic exponents $m$ and $n$). The linear diffusion term $D\nabla^2 z$ describes the rate of erosion or infilling by hillslope sediment transport processes. Finally, the uplift term $U$ gives the rate of tectonic uplift within the model domain (or, equivalently, base level fall at its boundary).

Equation (1) is defined piecewise on two subdomains. The first subdomain, where $\sqrt{A}|\nabla z| \leq \theta$, corresponds to areas where

the rate of incision is zero because it is fully suppressed by the incision threshold and, thus, the landscape evolves under the influence of diffusion and uplift only. We refer to these areas as the zones of zero incision, because they tend to form zones along ridges and drainage divides, where drainage area or slope, or both, are small (e.g., see Figs. 3 and 7). The second subdomain, where $\sqrt{A}|\nabla z| > \theta$, corresponds to the remaining parts of the landscape. In this subdomain, the incision rate is reduced by a uniform amount $K\theta$, relative to the rate $K\sqrt{A}|\nabla z|$ that would prevail with no threshold. Note that the transition

between the two subdomains at $\sqrt{A}|\nabla z| = \theta$ entails no discontinuity in incision rates, and that if we set $\theta = 0$, then we obtain the governing equation of the LEM without an incision threshold (e.g., Howard, 1994),

$$\frac{\partial z}{\partial t} = -K\sqrt{A}|\nabla z| + D\nabla^2 z + U \quad , \tag{2}$$

which is the equation that we dimensionally analyzed in Theodoratos et al. (2018).

### 2.2 Dimensions and characteristic scales

We test whether the characteristic scales defined in Theodoratos et al. (2018) are reasonable choices to analyze the LEM that

includes an incision threshold $\theta$ (Eq. 1). We start by examining the dimensions of the variables and parameters of Eq. (1). The horizontal coordinates $(x, y)$ have dimensions of length L, elevation $z$ has dimensions of height H, and time $t$ has dimensions of time T. These fundamental dimensions flow through to the derived topographic quantities: the rate of elevation change $\partial z/\partial t$ has dimensions of H T$^{-1}$, the drainage area $A$ has dimensions of L$^2$, the slope $|\nabla z|$ has dimensions of



H $L^{-1}$, and the Laplacian curvature $\nabla^2 z$ has dimensions of H $L^{-2}$. The fundamental dimensions are also applied to the model parameters: the incision coefficient $K$ has dimensions of $T^{-1}$, the diffusion coefficient $D$ has dimensions of $L^2 T^{-1}$, and the uplift rate $U$ has dimensions of H $T^{-1}$. In Eq. (1), the incision threshold $\theta$ is subtracted from the product $\sqrt{A}|\nabla z|$; thus, $\theta$ must have the same dimensions as $\sqrt{A}|\nabla z|$, i.e., dimensions of H. Note that in the more general case where the drainage area

and slope exponents $m$ and $n$ are not 0.5 and 1, respectively, $\theta$ must have the same dimensions as the product $A^m(|\nabla z|)^n$, i.e., $L^{2m-n} H^n$. LEMs typically express length and elevation in units of meters (m), and time in units of years (a). We also use these units in the simulations presented in Sect. 3, and we show incision coefficients $K$ in units of $a^{-1}$, diffusion coefficients $D$ in units of $m^2 a^{-1}$, uplift rates $U$ in units of m $a^{-1}$, and incision thresholds $\theta$ in units of m (e.g., Tables 1 and 2).

Given that all the terms of Eq. (1) have dimensions in H, L, and T, we can dimensionally analyze Eq. (1) using characteristic scales of length, height, and time. In Eqs. (3)-(8) below, we summarize the dimensional analysis of Theodoratos et al. (2018) as necessary background for the new analysis presented here. The dimensional analysis of Theodoratos et al. (2018) is based on a characteristic length that is defined as

$$l_c := \sqrt{D/K} \quad , \qquad\qquad\qquad\qquad\qquad\qquad\qquad\qquad\qquad\qquad (3)$$

a characteristic height that is defined as

$$h_c := U/K \quad , \qquad\qquad\qquad\qquad\qquad\qquad\qquad\qquad\qquad\qquad (4)$$

and a characteristic time that is defined as

$$t_c := 1/K \quad . \qquad\qquad\qquad\qquad\qquad\qquad\qquad\qquad\qquad\qquad (5)$$

To non-dimensionalize the horizontal coordinates $(x, y)$, elevation $z$, and time $t$, we divide them by $l_c$, $h_c$, and $t_c$, respectively. Specifically, we define dimensionless coordinates as $(x^*, y^*) := (x/l_c, y/l_c)$, dimensionless elevation as $z^* := z/h_c$, and dimensionless time as $t^* := t/t_c$.

To non-dimensionalize the rate of elevation change in the left-hand side of Eq. (1) and the topographic properties in the right-hand side of Eq. (1) (drainage area $A$, slope $|\nabla z|$, and curvature $\nabla^2 z$), we need to divide them by characteristic scales that have the same dimensions as the rate of elevation change and the topographic properties, respectively. We define these characteristic scales by combining the characteristic scales of length, height, and time ($l_c$, $h_c$, and $t_c$) such that we obtain the

appropriate dimensions. For instance, the rate of elevation change $\partial z/\partial t$ has dimensions of H $T^{-1}$; therefore, to non-dimensionalize it, we need to divide it by a characteristic scale with dimensions of height over time. The ratio $h_c/t_c$ has such dimensions. Note that $h_c/t_c = U$ (see Eqs. 4 and 5). Thus, we can view the uplift rate $U$ as a characteristic rate of elevation change and use it to define the dimensionless rate of elevation change as $\partial z^*/\partial t^* := (\partial z/\partial t)/U$. Likewise, we define a characteristic area with dimensions of $L^2$ as

$$A_c := l_c^2 = D/K \quad , \qquad\qquad\qquad\qquad\qquad\qquad\qquad\qquad\qquad (6)$$

and use it to define the dimensionless drainage area as $A^* := A/A_c$. Further, we define a characteristic gradient with dimensions of H $L^{-1}$ as

$$G_c := h_c/l_c = U/\sqrt{DK} \quad . \qquad\qquad\qquad\qquad\qquad\qquad\qquad\qquad (7)$$

If we divide the slope $|\nabla z|$ by the characteristic gradient $G_c$, we obtain a dimensionless slope term. We denote this dimensionless slope by $|\nabla^* z^*|$ because it is equal to the norm of the gradient of dimensionless elevation $z^*$ in dimensionless





coordinates $(x^*, y^*)$. Specifically, $|\nabla^* z^*| := \left|\frac{\partial z^*}{\partial x^*}\mathbf{i} + \frac{\partial z^*}{\partial y^*}\mathbf{j}\right| = \left|\frac{\partial z/h_c}{\partial x/l_c}\mathbf{i} + \frac{\partial z/h_c}{\partial y/l_c}\mathbf{j}\right| = \left|\frac{\partial z}{\partial x}\mathbf{i} + \frac{\partial z}{\partial y}\mathbf{j}\right|/(h_c/l_c) = |\nabla z|/G_c$. Finally, we

define a characteristic curvature with dimensions of H L$^{-2}$ as

$$\kappa_c := h_c/l_c^2 = U/D \quad , \tag{8}$$

and we use it to define the dimensionless curvature as $\nabla^{*2} z^* := \nabla^2 z/\kappa_c$. Note that the characteristic curvature is opposite to the steady-state curvature at ridges, drainage divides, and zones of zero incision. Specifically, if $\partial z/\partial t = 0$ (steady state),

and if $\sqrt{A}|\nabla z| = 0$ in Eq. (2) or $\sqrt{A}|\nabla z| \leq \theta$ in Eq. (1), then $D\nabla^2 z + U = 0$, which can be rewritten as $\nabla^2 z = -U/D = -\kappa_c$ (see also Roering et al., 2007; Perron et al., 2009). Note that it can be shown that $-\kappa_c$ is also the minimum value of curvature in steady state.

## 2.3 Dimensionless governing equation and the incision-threshold number $N_\theta$

If we divide all of the terms of the governing equation (Eq. 1) by the uplift rate $U$, then we obtain an equation that includes only dimensionless terms. Specifically, if we divide the left-hand side of Eq. (1) by $U$, then we obtain the dimensionless rate of elevation change $\partial z^*/\partial t^* := (\partial z/\partial t)/U$. In the right-hand side of Eq. (1), dividing the uplift term $U$ by itself yields the number 1. Furthermore, dividing the diffusion term $D\nabla^2 z$ by $U$ and substituting the characteristic curvature $\kappa_c$ (Eq. 8), we obtain $(D\nabla^2 z)/U = \nabla^2 z/(U/D) = \nabla^2 z/\kappa_c = \nabla^{*2} z^*$, which is the dimensionless form of the diffusion term in Eq. (1).

To non-dimensionalize the incision term, we expand it to $K\sqrt{A}|\nabla z| - K\theta$. The first part, $K\sqrt{A}|\nabla z|$, corresponds to the value of the incision rate if there were no incision threshold, and the second part, $K\theta$, corresponds to the reduction in the incision rate due to the threshold. If we divide $K\sqrt{A}|\nabla z|$ by $U$, then it can be shown that we obtain the dimensionless product $\sqrt{A^*}|\nabla^* z^*|$, and if we divide $K\theta$ by $U$, then we obtain $K\theta/U$. This ratio is dimensionless because both $K\theta$ and $U$ are rates of

elevation change, with dimensions of H T$^{-1}$. We term this dimensionless ratio the incision-threshold number $N_\theta$:

$$N_\theta := K\theta/U \quad . \tag{9}$$

The incision-threshold number $N_\theta$ quantifies $K\theta$, the reduction in the rate of incision due to the incision threshold, relative to the uplift rate $U$. Additionally, $N_\theta$ is the value of the dimensionless product $\sqrt{A^*}|\nabla^* z^*|$ at the transition between the two subdomains of Eq. (1), i.e., at the interface between parts of the landscape where there is no incision and parts of the

landscape where incision occurs. Specifically, at that transition, $\sqrt{A}|\nabla z| = \theta$; if both sides of this equality are multiplied by $K$ and then divided by $U$, then the equality can be shown to become $\sqrt{A^*}|\nabla^* z^*| = N_\theta$. Finally, if we rearrange Eq. (9) as $N_\theta = \theta/(U/K)$, then we see that the incision-threshold number gives the magnitude of the incision threshold $\theta$ relative to magnitudes of other parameters of the LEM, specifically, relative to the ratio of the uplift rate $U$ to the incision coefficient $K$. Note that in the general case in which the drainage area and slope exponents $m$ and $n$ are not 0.5 and 1, respectively, $\theta$ and $K$

will have different dimensions than in the case of Eq. (1), but their product, $K\theta$, will still have dimensions of H T$^{-1}$. Thus the ratio $K\theta/U$ is dimensionless for any $m$ and $n$.

Bringing together the dimensionless terms derived above, we obtain a dimensionless form of the governing equation (Eq. 1):

$$\frac{\partial z^*}{\partial t^*} = \begin{cases} \nabla^{*2} z^* + 1, & \sqrt{A^*}|\nabla^* z^*| \leq N_\theta \\ -\left(\sqrt{A^*}|\nabla^* z^*| - N_\theta\right) + \nabla^{*2} z^* + 1, & \sqrt{A^*}|\nabla^* z^*| > N_\theta \end{cases} \quad . \tag{10}$$





Note that the dimensionless quantities that we denote by starred symbols (e.g., $z^*$, $A^*$, $|\nabla^* z^*|$) refer to variables or topographic properties. These quantities vary in space across the landscape and in time as the landscape evolves. By contrast, the incision-threshold number $N_\theta$ depends only on the model parameters $K$, $U$, and $\theta$, and thus it plays the role of a parameter in Eq. (10), one that is constant in space and time. The incision-threshold number $N_\theta$ is the only parameter in Eq. (10). Thus, 5 for a given set of boundary and initial conditions, the value of $N_\theta$ is the only control on the solution of Eq. (10).

The LEM without incision threshold, which we studied in Theodoratos et al. (2018), has a dimensionless form that does not include any parameters (see Eq. (16) in Theodoratos et al., 2018). Having no parameters to be adjusted, the dimensionless form has a single solution for any given combination of boundary and initial conditions. This implies that landscapes with 10 any parameters, but with the same boundary and initial conditions (when normalized by the characteristic scales $l_c$ and $h_c$), follow geometrically similar evolutionary paths, i.e., they evolve as rescaled copies of each other. We noted that this rescaling property implies that, instead of running multiple simulations corresponding to multiple combinations of parameters, we can explore the entire parameter space of the LEM by rescaling the results of a single simulation corresponding to just one set of parameters.

In contrast, the dimensionless form of the LEM with an incision threshold, Eq. (10), includes one parameter, the incision-threshold number $N_\theta$. Therefore, in general, landscapes with non-zero incision thresholds will not evolve as rescaled copies. However, Eq. (10) reveals a special case. If landscapes have the same $N_\theta$, i.e., if they have incision thresholds $\theta$, incision coefficients $K$, and uplift rates $U$ such that they have the same ratios $K\theta/U$, then they will evolve as rescaled copies of each 20 other, provided that their boundary and initial conditions are the same when normalized by the characteristic scales of length and height $l_c$ and $h_c$. In Sect. 3, we numerically demonstrate both the special case of landscapes that have the same $N_\theta$ and evolve geometrically similarly, and the general case of landscapes that have different $N_\theta$ and do not follow geometric similarity.

25 The elimination of three out of four parameter-related degrees of freedom from the LEM (from the four parameters $K$, $D$, $U$, and $\theta$ in Eq. (1) to the one parameter $N_\theta$ in Eq. (10)) is a substantial simplification. It is a consequence of the fact that we non-dimensionalize Eq. (1) using the characteristic scales of length, height, and time $l_c$, $h_c$, and $t_c$, which depend on three model parameters ($K$, $D$, and $U$; Eqs. 3–5), and can thus eliminate an equal number of parameter-related degrees of freedom. This simplification validates the hypothesis that $l_c$, $h_c$, and $t_c$, as a group, remain useful in the case of Eq. (1), which includes 30 the incision threshold $\theta$. Unfortunately, with only three fundamental dimensions it is not possible to eliminate all four parameters using dimensional analysis, so one dimensionless parameter (in this case $N_\theta$) must remain.

## 3 Numerical simulations

### 3.1 Special case: landscapes with the same $N_\theta$

In this section, we numerically demonstrate that landscapes that follow Eq. (1) but have different parameters will evolve 35 geometrically similarly if they have equal incision-threshold numbers $N_\theta$, provided that their boundary and initial conditions are equivalent when normalized by the characteristic scales of length and height $l_c$ and $h_c$. Given that we perform numerical





simulations on discrete and finite domains, we also normalize the sizes and resolutions of these domains by $l_c$ (see Sects. 2.2 and 3.2.2 of Theodoratos et al. (2018) for a more detailed discussion regarding the rescaling of domain size and resolution).

In this context, geometric similarity is defined in the following way. Let the first landscape have characteristic scales $l_c$ and
$h_c$, and the second have $l_c'$ and $h_c'$. The two landscapes are geometrically similar if any point with coordinates $(x, y)$ and elevation $z$ from the first landscape corresponds to a point from the second landscape with coordinates $(x', y')$ and elevation $z'$ such that $(x/l_c, y/l_c) = (x'/l_c', y'/l_c')$ and $z/h_c = z'/h_c'$. Note that both points correspond to the same point of a dimensionless landscape with coordinates $(x^*, y^*) = (x/l_c, y/l_c) = (x'/l_c', y'/l_c')$ and elevation $z^* = z/h_c = z'/h_c'$. In other words, the two landscapes are geometrically similar if they correspond to the same dimensionless landscape. To test
whether the two landscapes are geometrically similar during their evolution, we must normalize time by their characteristic timescales $t_c$ and $t_c'$. Specifically, we must compare a snapshot of the first landscape at some time $t$ to a snapshot of the second landscape at some other time $t'$, such that $t/t_c = t'/t_c'$. Both of these snapshots correspond to the same snapshot of a dimensionless landscape at a dimensionless time $t^* = t/t_c = t'/t_c'$.

### 3.1.1 Setup of simulations

We perform numerical simulations using the Channel-Hillslope Integrated Landscape Development (CHILD) model (Tucker et al., 2001). Below, we briefly explain how we set up the simulations, and in Appendix A we present formulas that relate the parameters of CHILD to the parameters of the governing equation (Eq. 1). We refer readers to Theodoratos et al. (2018) for more details about setting up numerical simulations that follow geometric similarity (Sect. 3.1.1 and Appendix C) and about the theory behind such simulations (Appendix B).

For our similarity analysis, we simulate nine landscapes, each having a different combination of the parameters $K$, $D$, and $U$, and, thus, a different combination of characteristic scales of length and height $l_c$ and $h_c$ (Eqs. 3, 4). Using Eq. (9), we determine the value of the incision threshold $\theta$ of each landscape such that the incision-threshold number of all landscapes is $N_\theta = 1$. We show the parameters, characteristic scales, and $\theta$ and $N_\theta$ values of the nine landscapes in Table 1. The
landscapes are named with capital letters, from A to I.

Note that the incision threshold values $\theta$ of some of the nine landscapes are significantly higher than natural values reported in the literature (e.g., Prosser & Dietrich, 1995; note the necessary unit conversions). This is due to the fact that all nine landscapes have incision-threshold numbers $N_\theta = \theta/(U/K) = 1$, i.e., due to the fact that each landscape's $\theta$ value must be
equal to the value of its $U/K$ ratio. We chose to use the value $N_\theta = 1$ because it leads to wide zones of zero incision (areas where, according to Eq. (1), there is no incision, because $\sqrt{A}|\nabla z| \leq \theta$). These wide zones are readily visible when plotted.

To obtain domains and initial conditions that are equivalent when normalized by the characteristic scales of length and height $l_c$ and $h_c$, we first synthesize a random triangular irregular network (TIN) in dimensionless space, i.e., a TIN whose
vertices have dimensionless horizontal coordinates $(x^*, y^*)$ and dimensionless initial elevations $z^*$. (This TIN's total extent is 60×90, the average length of its triangle edges is 0.4, and the initial elevations are a white noise ranging between 0 and 0.1.) Second, we multiply $(x^*, y^*)$ and $z^*$ by each landscape's $l_c$ and $h_c$, respectively. Thus we obtain each landscape's dimensional TIN with horizontal coordinates $(x, y)$ and initial elevations $z$.



Note that landscapes can reach geometrically similar steady states only if the criteria that define the steady state are normalized by appropriate characteristic scales, as explained in Sect. 3 of Theodoratos et al. (2018). In the present study, for instance, we assume that a simulation reaches its steady state when the absolute rate of elevation change $|\partial z/\partial t|$ falls below a limit $\varepsilon$ at all points. Given that $\varepsilon$ is a rate of elevation change, we can normalize it by the uplift rate $U$, which can be viewed as a characteristic rate of elevation change, as we explain in Sect. 2.2. Thus, we set each simulation's limit to $\varepsilon = 0.001\,U$.

### 3.1.2 Results: geometric similarity

The nine simulated landscapes are all geometrically similar to each other, both during their evolution and in steady state. In Figs. 1–3, we graphically demonstrate that our simulated landscapes reach geometrically similar steady states. Specifically, we illustrate shaded relief maps in Fig. 1, elevation maps in Fig. 2, and maps of the extents of the zones of zero incision in Fig. 3. (In the present study, we illustrate only steady-state results. For examples of graphical demonstrations of geometric similarity during landscape evolution, we refer readers to Figs. 3–5 of Theodoratos et al., 2018.) For clarity, we present maps of only four out of the nine landscapes, specifically, of landscapes A–D in Table 1. However, all nine landscapes evolve geometrically similarly.

In Figs. 1–3, the four landscapes are arranged in a 2x2 array, such that the incision threshold $\theta$ increases from top to bottom and from left to right. The characteristic height $h_c$ follows the same arrangement as $\theta$, because $h_c = U/K = \theta/N_\theta$ and all landscapes have the same $N_\theta$. The characteristic length $l_c$ increases independently of $h_c$ and $\theta$, specifically, from bottom to top and from left to right. The coloring and labeling of Figs. 1–3 highlight both the large differences of scale and the geometric similarity of the four landscapes. Specifically, lengths and elevations on axes and colorbars are shown both in units of km or m using bold fonts, and in units of $l_c$ or $h_c$ using normal fonts. Further, color scales of elevation maps in Fig. 2 are rescaled by $h_c$ to assist with comparing the elevations of features. Note that a quantity shown in units of the corresponding characteristic scale has the same numerical value as the dimensionless version of this quantity, e.g., elevation $z$ in units of $h_c$ has the same numerical value as dimensionless elevation $z^*$ because both values are given by the formula $z/h_c$. Therefore, in Figs. 1–3, the values of quantities shown in units of $l_c$ or $h_c$ with normal fonts are the same as the values of the corresponding dimensionless quantities.

In the shaded relief maps of Fig. 1, ridges and valleys form identical plan-view patterns across the four landscapes, illustrating their horizontal geometric similarity. Note that the characteristic scales of length and height $l_c$ and $h_c$ vary independently, leading to different characteristic gradients $G_c$ across the landscapes. Therefore, landscape features in different landscapes have different steepness and, thus, they are shown with different shades of gray.

In the elevation maps of Fig. 2, the spatial pattern of colors is identical across the four landscapes. This shows that the four landscapes are geometrically similar both horizontally and vertically, because the color scales are rescaled by $h_c$.

In Fig. 3, we map the zones of zero incision of the four landscapes. To illustrate these zones, we find the Voronoi polygons associated with points for which $\sqrt{A}|\nabla z| \leq \theta$ and we color them red. (Each point of the simulated landscapes is a TIN vertex. The associated Voronoi polygon is the area that is assumed to drain to that point; see Tucker et al., 2001.) We



observe that the spatial patterns of the red Voronoi polygons in all four maps are geometrically similar. This implies that the zones of zero incision of the four landscapes have geometrically similar horizontal extents.

The landscapes in Figs. 1–3 do not just visually appear to be geometrically similar. They are in fact geometrically similar. To test this quantitatively, we normalize the elevations $z$ of each landscape by its characteristic height $h_c$ and compare the resulting dimensionless elevations $z^* = z/h_c$ of different landscapes. As we explain further above, the dimensionless elevations $z^*$ of geometrically similar landscapes must be equal. Indeed, for the nine landscapes of Table 1, we find that the maximum absolute difference between steady-state $z^*$ values of corresponding points is less than $3 \times 10^{-9}$.

### 3.2 General case: landscapes with different $N_\theta$

In this subsection, we demonstrate that landscapes with different incision-threshold numbers $N_\theta$ do not evolve geometrically similarly, even if their domains and initial conditions are rescaled by the characteristic scales of length and height $l_c$ and $h_c$. Further, we illustrate how the differences in the value of $N_\theta$ are reflected in the topography of these landscapes.

#### 3.2.1 Setup of simulations

For these simulations, too, we use CHILD, as described in Appendix A. We perform nine simulations with incision-threshold numbers $N_\theta$ that range between 0 and 4. We use a single combination of values for the incision coefficient $K$, diffusion coefficient $D$, and uplift rate $U$, and we vary the incision-threshold number $N_\theta$ by varying only the incision threshold $\theta$ (see Eq. 9). Therefore, all nine simulations have the same characteristic scales of length and height (specifically, $l_c$=50 m and $h_c$=25 m). Thus, for all nine simulations, we use the same domains and initial conditions. Specifically, we use TINs with total extent of 150 $l_c$ × 225 $l_c$ (i.e., 7.5 km × 11.25 km), average TIN edge length of 0.4 $l_c$ (i.e., 20 m), and random initial elevations drawn from a uniform distribution ranging between 0 and 0.1. The parameters $K$, $D$, and $U$ have values that fall within the typical range seen in the literature (e.g., Perron et al., 2008; Tucker, 2009). In contrast, the incision coefficients $\theta$ that correspond to the highest values of $N_\theta$ that we examine here have values that far exceed real-world incision threshold values typically reported (e.g., Prosser & Dietrich, 1995; note the necessary unit conversions). However, we use these high values to examine how the LEM behaves when $N_\theta$ is high. The values of $K$, $D$, and $U$, and of $\theta$ and the corresponding $N_\theta$ of the nine landscapes are shown in Table 2.

#### 3.2.2 Results: lack of geometric similarity and illustration of influence of $N_\theta$ on landscape topography

As we mentioned in the Introduction (Sect. 1), the inclusion of incision thresholds in LEMs leads to increasing topographic slopes, decreasing drainage densities, and more convex hillslopes (e.g., Howard, 1994; Tucker and Bras, 1998; Perron et al., 2008). In Figs. 4–10, we illustrate these topographic effects using steady-state results of the nine simulations defined above (Sect. 3.2.1, Table 2). More specifically, we present shaded relief maps (Figs. 4 and 5), maps of elevation $z$ (Fig. 6), maps of the extents of the zones of zero incision (Fig. 7), maps of curvature $\nabla^2 z$ (Fig. 8), and profiles from ridge to outlet along flow paths (Figs. 9, 10). We show profiles along each landscape's longest flow path to make profiles of different landscapes comparable. We mark these flow paths with blue lines on the maps of Figs. 4–8. The maps in Figs. 4 and 6 show the full extent of the landscapes, which is 7.5 km × 11.25 km (i.e., 150 $l_c$ × 225 $l_c$), whereas the maps in Figs. 5, 7, and 8 show magnified versions of a 5 km × 4 km (i.e., 100 $l_c$ × 80 $l_c$) rectangular region from each map. To make the regions of different landscapes comparable, we select each region such that it contains the drainage basin of the longest flow path of





each landscape. We mark these regions with blue rectangles in Figs. 4 and 6. Note that, in all of these figures, we show quantities in units of m (or km in the case of horizontal lengths) using bold fonts and in units of the corresponding characteristic scales using normal fonts (which yield the same numerical values as dimensionless versions of quantities, as we explain in Sect. 3.1.2). Likewise, we show each simulation's incision threshold $\theta$ (in units of m) using bold fonts and the

corresponding incision-threshold number $N_\theta$ (dimensionless) using normal fonts.

We observe that landscapes become steeper as $N_\theta$ increases. Specifically, in the shaded relief maps (Figs. 4, 5), hillslopes are shown with darker shades of gray, i.e., they become steeper, and in the profile plots (Fig. 9), the landscapes' longest flow paths become steeper. Given that all landscapes have the same horizontal extents, the steepening of landscapes implies that

landscape relief increases. We observe the increase of relief with $N_\theta$ both in terms of the maximum value of elevation (see labels at the bottom of elevation maps in Fig. 6) and in terms of the whole distribution of elevation (see profiles in Fig. 9 and the range of colors of elevation maps in Fig. 6).

Furthermore, we observe that landscapes become less dissected and appear smoother in plan view as $N_\theta$ increases.

Specifically, in the shaded relief maps (Figs. 4, 5), we see that the smooth, undissected areas along the sides of ridges and interfluves become wider, and the tips of valley networks move away from the ridges. In the maps of curvature (Fig. 8), we see that the valley networks become sparser, i.e., the landscapes become less dissected. For the case of valley heads that fall on the landscapes' longest flow paths, we see the movement away from the ridges also in the profile plots of Fig. 9 (see blue circles).

We observe that, as $N_\theta$ increases, valleys become deeper (more concave). Specifically, in the maps of curvature (Fig. 8), the maximum value of curvature increases with $N_\theta$ and, thus, the positive values of curvature become more positive. In other words, concave areas, which can be defined as valleys (e.g., Howard, 1994), become more concave. Additionally, in the shaded relief maps (Figs. 4, 5), valleys in landscapes with higher $N_\theta$ appear deeper because their contrast with neighboring

hillslopes is higher. Note that the deepening of valleys is in agreement with the steepening of hillslopes described above.

Moreover, we observe that as $N_\theta$ increases, the zones of zero incision (i.e., the areas with $\sqrt{A}|\nabla z| \leq \theta$; shown with red in Fig. 7) become wider and occupy bigger portions of the hillslopes. We can also observe this in the profile plots of Fig. 9. Specifically, we see that, as $N_\theta$ increases, the red dots move away from the ridge and come closer to the blue circles, which

implies that the longest flow paths' segments that have zero incision become longer and that they occupy bigger portions of the segments that belong to hillslopes.

Consequently, hillslopes become more convex as $N_\theta$ increases. In steady state, the curvature in zero-incision zones is equal to $-\kappa_c$ (the negative of the characteristic curvature), which is the minimum value of curvature (see Sect. 2.2). Thus, the

widening of zero-incision zones implies that bigger portions of hillslopes acquire the minimum curvature, i.e., bigger portions of them become maximally convex. (Note, however, that the value of the maximum convexity remains constant as $N_\theta$ increases, because the minimum curvature remains $\nabla^2 z = -\kappa_c$.) The maps of curvature (Fig. 8) also show that the minimum value of curvature remains constant as $N_\theta$ increases.



Finally, we observe that the widening of the zones of zero incision eventually leads to a qualitative change in the operation of the laws of the LEM across the landscapes. Specifically, the zones of zero incision almost entirely occupy the hillslopes of the landscape with $N_\theta = 4$. We deduce this by observing in Fig. 7 that the white areas (i.e., areas with $\sqrt{A}|\nabla z| > \theta$, where incision does operate) follow the pattern of the dendritic valley network. The almost complete occupation of hillslopes by the

zones of zero incision implies that incision operates almost exclusively in valleys, which is a qualitative change. The governing equation without incision threshold (Eq. 2) is based on the fundamental assumption that all of its processes (incision, diffusion, and uplift) operate everywhere across a landscape (e.g., Howard, 1994). By including the incision threshold $\theta$, the governing equation Eq. (1) becomes piecewise, with a first subdomain with $\sqrt{A}|\nabla z| \leq \theta$ where only diffusion and uplift operate, and a second subdomain with $\sqrt{A}|\nabla z| > \theta$ where all three processes operate. This formulation

does not exclude incision from hillslopes in principle. In effect, however, incision is excluded from hillslopes for high values of the incision-threshold number $N_\theta$, as revealed by the white dendritic patterns in Fig. 7. Thus, for $N_\theta = 4$ the governing equation (Eq. 1) is, in effect, reminiscent of LEMs that explicitly define distinct laws for hillslopes and valleys (e.g., Willgoose et al., 1991; Goren et al., 2014). Note that increasing $N_\theta$ beyond the value of 4 would not lead to the additional qualitative change of zero-incision zones starting to occupy valleys, because zero-incision zones have negative curvature

($\nabla^2 z = -\kappa_c$; see Sect. 2.2). Note that $N_\theta = 4$ is the value for which hillslopes are completely occupied by zero-incision zones in the landscapes that we examine here. However, in landscapes with different boundary and initial conditions, the qualitative change described here could occur at different values of the incision threshold number $N_\theta$.

With the above observations in mind, we can explain the observation that landscapes become steeper as $N_\theta$ increases in two

distinct ways, one referring to areas outside zero-incision zones and one referring to areas within them. First, channels become steeper to compensate for the reduction in the strength of incision by the incision threshold. Equation (1) shows that incision operates in areas with $\sqrt{A}|\nabla z| > \theta$, but the rate of incision is reduced by the quantity $K\theta$ relative to $K\sqrt{A}|\nabla z|$, which is the rate of incision in a landscape without incision threshold. Therefore, for a given drainage area $A$, the landscape must have steeper slope $|\nabla z|$ to achieve the same incision rate, and thus balance the other processes and reach equilibrium. This

effect becomes stronger as $N_\theta$ increases. Second, for purely geometrical reasons, the fact that hillslopes become more convex as $N_\theta$ increases implies that they also become steeper. Typically, the more negative the Laplacian curvature $\nabla^2 z$ of an area, the faster is the increase of slope over a given flow path length. (Exceptions can be areas with negative contour curvature, but positive profile curvature, where slope decreases along flow paths, e.g., wind gaps; see also Fig. 2, panel (c) in Mitasova and Hofierka, 1993.) Therefore, as $N_\theta$ increases and hillslopes become more convex, the slope at a given distance from the

ridge becomes steeper.

In an alternative interpretation, one could potentially view the quantity $K\theta$ not as a reduction of the rate of incision, but rather as a virtual source term, i.e., as a virtual increase of the uplift rate $U$. Thus the observed increase of relief would be interpreted as resulting from the virtual increase of the uplift rate because, all else remaining equal, higher uplift rates lead to

higher reliefs (e.g., Tucker and Whipple, 2002; Theodoratos et al., 2018). However, this view is not meaningful in the zones of zero incision, because in the first subdomain of Eq. (1) the quantity $K\theta$ does not appear and, thus, $U$ is the only source term (this is also reflected in the fact that ridgelines do not become more sharply convex as they would if the uplift rate were actually increased; rather, the curvature of ridgelines remains equal to $-\kappa_c = -U/D$). To quantify how the uplift rate's virtual increase depends on the incision-threshold number $N_\theta$, we can rearrange the right-hand side of the second subdomain



of the governing equation (Eq. 1). We take the quantity $K\theta$ from the incision term and we group it with the uplift rate $U$. Thus, we form the virtual uplift rate $K\theta + U$, which we rewrite as

$$K\theta + U = (U/U) \cdot K\theta + U = (K\theta/U + 1)U = (N_\theta + 1)U \quad . \tag{11}$$

Because Eq. (11) does not apply within the zones of zero incision, treating $K\theta$ as a virtual increase of the uplift rate implies that one must also treat the landscape as having two distinct uplift rates, $(N_\theta + 1)U$ outside the zones of zero incision and $U$

within them.

Equation (11) suggests that the quantity $N_\theta + 1$ can predict how the relief of a landscape (outside the zones of zero incision) depends on the value of the incision-threshold number $N_\theta$. All else being equal, relief is proportional to the uplift rate (e.g., see definition of the uplift erosion number $N_E$ in Tucker and Whipple, 2002, or interpretations of our characteristic height $h_c$

in Theodoratos et al., 2018). Therefore, Eq. (11) suggests that relief (outside zero-incision zones) is proportional to $N_\theta + 1$ (because the virtually increased uplift rate is proportional to $N_\theta + 1$), implying that elevations (outside zero-incision zones) in landscapes that differ only in their $N_\theta$ values would be equal when normalized by $N_\theta + 1$.

We can test this hypothesis using the profiles of Fig. 9, since they belong to landscapes that have different incision-threshold

numbers $N_\theta$, but the same parameters, characteristic scales, domains, and initial conditions (see Table 2). Specifically, we divide elevations along each profile of Fig. 9 by $N_\theta + 1$, and we plot the resulting normalized profiles in Fig. 10. The hypothesis will not be rejected if the normalized profiles have the same normalized elevations outside the zones of zero incision. Indeed, we observe that the normalized elevations are nearly equal, especially in those reaches of each profile that are not near its zone of zero incision. This suggests that, away from the zero-incision zones, landscape relief nearly scales

with $N_\theta + 1$.

In Fig. 10, we observe that the elevations of the normalized profiles deviate systematically from one another. Specifically, we observe that, whereas the reliefs of the original (un-normalized) profiles grow as $N_\theta$ increases, the reliefs of the normalized profiles decrease as $N_\theta$ increases. (In Table 3 we show an example of this reversal using the original and

normalized elevations of the profiles at a distance of 0.5 km from the ridge, which falls outside the zones of zero incision of all profiles; see arrows in Fig. 10.) This reversal implies that normalizing elevations by $N_\theta + 1$ is an overshoot, as it lowers the profiles by a larger factor than what would make them equal to each other. In other words, as $N_\theta$ increases, the elevation of the original profiles is increased less than proportionally to $N_\theta + 1$, i.e., less than what is predicted by viewing the quantity $K\theta$ as a virtual increase of the uplift rate. This suggests that the incision threshold could be resulting in additional

effects, which oppose the virtual increase of the uplift rate, and that these effects depend on the value of $N_\theta$. Future work can study such effects. For example, it is known that incision's competition with diffusion for the propagation of elevation perturbations can be influenced by the incision threshold (e.g., see relationship between the Péclet number and the incision threshold in Perron et al., 2008). Thus it may be productive to examine whether changes in the competition between incision and diffusion alter how the incision threshold affects the rate of incision.





## 4 Discussion

### 4.1 On the definition of zones of zero incision

Unlike the LEM studied here, other LEMs, such as those of Tucker (2004) or Deal et al. (2018), do not define zones of zero incision, i.e., areas where incision never operates, because those LEMs define incision terms based on conceptually different
temporal averaging of rainfall events, in comparison to the LEM examined here.

Specifically, those other LEMs derive long-term incision rates by integrating stochastic rainfall over time, assuming that incision occurs when the shear stress (or, equivalently, the stream power) exceeds a threshold value. Given that the value of shear stress depends on discharge and slope, points with different drainage areas or slopes will experience different shear
stress values during any given event. Therefore, any given combination of drainage area and slope corresponds to a critical rainfall intensity that is sufficient to generate a shear stress that equals the threshold shear stress. Long-term incision rates can be derived by integrating over the rainfall events that exceed this critical rainfall intensity. This approach implies that, in theory, any point with non-zero drainage area and slope can experience incision with a non-zero probability (provided that rainfall can theoretically become sufficiently intense). Therefore, in LEMs that follow this approach, zero-incision zones are
not defined. (Note, however, that in those LEMs one can define zones of probability of exceedance of the critical rainfall intensity, i.e., of probability of incision.)

In contrast, the LEM studied here assumes constant, uniform rainfall, which leads to constant stream power for any given combination of drainage area and slope (i.e., for any given value of $\sqrt{A}|\nabla z|$). Thus, points with any given $\sqrt{A}|\nabla z|$ either
experience stream power above the threshold value, which leads to incision, or they do not. Therefore, instead of explicitly including a stream-power threshold, the LEM's governing equation (Eq. 1) uses a threshold of the quantity $\sqrt{A}|\nabla z|$ itself, namely, the incision threshold $\theta$ (see the relationship between $\theta$ and the threshold of stream power in Appendix A). This formulation of Eq. (1) has the advantage of being much simpler than those of LEMs that use stochastic rainfall and shear-stress (or stream-power) thresholds. However, Eq. (1) has the disadvantage of being unable to explore the non-linear
relationship between average precipitation and long-term incision rates that we describe in the Introduction (Sect. 1).

### 4.2 On the choice of characteristic scales

In this study, we have examined whether the characteristic scales of length, height, and time ($l_c$, $h_c$, and $t_c$; Eqs. 3–5), which we introduced in Theodoratos et al. (2018), remain useful after the inclusion of an incision threshold in the LEM, and we find that they do. Furthermore, while non-dimensionalizing Eq. (1) using this group of characteristic scales, we obtain the
dimensionless incision-threshold number $N_\theta$, which has useful properties. These results, however, do not imply that $l_c$, $h_c$, and $t_c$ are the only possible choices of characteristic scales, or even that they are the best choices for all problems. For any given model, different characteristic scales may be more appropriate for different applications.

Dimensional analysis can ensure that a set of characteristic scales is dimensionally consistent and can provide the number of
degrees of freedom that can be eliminated from a model (e.g., Buckingham, 1914), but it cannot show a priori which characteristic scales should be used. For example, in the case of Eq. (1), if we assume that length L and height H are distinct dimensions, then together with time T they form a group of three dimensions, and dimensional analysis will show that any





manipulation of Eq. (1) can eliminate at most 3 degrees of freedom. Because the characteristic scales $l_c$, $h_c$, and $t_c$ are defined by the parameters $K$, $D$, and $U$, eliminating three degrees of freedom eliminates these three parameters. If, instead, one defined characteristic scales that depended, for example, on the measurements of the domain (e.g., Perron et al., 2008), the corresponding degrees of freedom that could be eliminated would be related to these domain scales. Such an approach might

be more appropriate for characterizing extensive properties of a landscape as a whole (e.g., Perron et al., 2012), whereas the approach that we use here may be more appropriate for characterizing processes and intensive properties that vary across a landscape (e.g., Theodoratos et al., 2018). It may be difficult to predict a priori which choices of characteristic scales will be better for a given problem, and the only way to find out may be to try several different alternatives. In general, dimensional analysis can be used to simplify governing equations, and it can point to useful numerical, field, or lab experiments, but it

cannot fully substitute the information contained in empirical results (e.g., Huntley, 1967).

## 5    Summary and conclusions

In this study, we perform a dimensional analysis of an LEM that includes terms describing stream-power incision, linear diffusion, and uplift (Eq. 1). The LEM assumes that incision is limited by a threshold, specifically, that there is no incision at points with drainage area $A$ and slope $|\nabla z|$ such that the quantity $\sqrt{A}|\nabla z|$ is below a threshold value $\theta$, and that this threshold

also reduces incision at points with $\sqrt{A}|\nabla z| > \theta$.

Our dimensional analysis is based on characteristic scales of length, height, and time ($l_c$, $h_c$, and $t_c$) that depend only on parameters of the LEM (specifically, on the incision coefficient $K$, diffusion coefficient $D$, and uplift rate $U$; Eqs. 3–5). We introduced these scales as a group in Theodoratos et al. (2018), where we analyzed a related LEM that did not include the

incision threshold (reproduced here as Eq. 2). The distinction between $l_c$ and $h_c$ is based on the assumption that horizontal lengths and vertical heights are dimensionally distinct.

In Sect. 2.3, using the characteristic scales $l_c$, $h_c$, and $t_c$, we derive Eq. (10), a dimensionless form of the governing equation of the LEM that includes only one parameter, the incision-threshold number $N_\theta = K\theta/U = \theta/(U/K)$ (Eq. 9). This

dimensionless parameter quantifies the value of $K\theta$, which is the reduction in the rate of incision due to the incision threshold, relative to the uplift rate $U$ or, equivalently, the relative magnitude of the incision threshold $\theta$ versus the ratio $U/K$. The original, dimensional LEM (Eq. 1) includes four parameters ($K$, $D$, $U$, and $\theta$). Because the three characteristic scales ($l_c$, $h_c$, and $t_c$) depend on three model parameters ($K$, $D$, and $U$), in deriving the dimensionless Eq. (10) we can eliminate three out of four parameter-related degrees of freedom, which is a notable simplification. This suggests that this

group of characteristic scales is useful in the case of the LEM that includes an incision threshold.

The fact that the incision-threshold number $N_\theta$ is the only parameter in the dimensionless governing equation (Eq. 10) implies that it is the only control on this equation, for any given set of boundary and initial conditions. As a consequence, the evolution of all landscapes with a given $N_\theta$ value will be geometrically and temporally similar, provided that their domains,

boundary conditions, and initial conditions are rescaled by $l_c$ and $h_c$ (see Theodoratos et al. (2018) for more detailed theoretical exposition of these arguments). In Sect. 3.1, we present numerical simulations of landscapes with different parameters but equal incision-threshold numbers $N_\theta$. In Figs. 1–3, we demonstrate that these landscapes indeed evolve





geometrically similarly. In contrast, landscapes with different $N_\theta$ values evolve without geometric similarity, as we show with a second set of numerical simulations in Sect. 3.2.

In Sect. 3.2.2, we explore how these different $N_\theta$ values influence steady-state topographic properties of the resulting
landscapes. We illustrate the topographic influence of $N_\theta$ in Figs. 4–10. We find that, as $N_\theta$ increases, landscape relief increases (Figs. 6, 9) and, thus, hillslopes and channels become steeper (Figs. 4, 5, 9), valleys become sparser but also deeper (Figs. 4, 5, 8), and hillslopes become more convex (Figs. 7, 8).

Finally, we derive a quantitative prediction of the increase of relief with the incision-threshold numbers $N_\theta$. Specifically, we
show that in areas with $\sqrt{A}|\nabla z| > \theta$, where incision operates, relief tends to scale with the quantity $N_\theta + 1$ and thus elevations tend to become equal if they are normalized by $N_\theta + 1$ (Fig. 10). Our simulation results show deviations from this prediction, but we observe that these deviations are systematic (Sect. 3.2.2, Table 3) and we posit that the incision threshold causes additional effects which can be the focus of future work.

## Appendix A: Implementation of governing equation with CHILD

To implement the governing equation of the LEM (Eq. 1) with CHILD, we use CHILD's detachment-limited module and we set the parameter DETACHMENT_LAW equal to 0. Furthermore, we use constant, uniform, and continuous precipitation, we define infiltration to be 0, and we set the hydraulic geometry scaling exponents $\omega_b$ and $\omega_s$ to be equal to 0.5, and the detachment capacity exponents $m_b$, $n_b$, and $P_b$ to be equal to 1 (see Tucker et al., 2001, and Tucker, 2010, for definitions of CHILD's assumptions, modules, and parameters).

For this choice of exponents, CHILD uses the following equations to calculate the rate of elevation change due to incision (in CHILD notation):

$$\left.\frac{\partial z}{\partial t}\right|_{\text{Incision}} = -D_c = -k_b(\tau_0 - \tau_c) \quad , \tag{A1 a}$$

$$\tau_0 = k_t \frac{\sqrt{P}\sqrt{A}}{k_w} S \quad , \tag{A1 b}$$

where $D_c$ is the maximum detachment capacity in ma$^{-1}$, $\tau_0$ is stream power per unit bed area in W m$^{-2}$, $\tau_c$ is the threshold value of $\tau_0$, below which there is no incision, also in W m$^{-2}$, $k_b$ is a detachment rate coefficient in m a$^{-1}$ (W m$^{-2}$)$^{-1}$ (i.e., $k_b$ is
the rate of elevation change per each unit of stream power per unit bed area), $k_t$ is the specific weight of water in N m$^{-3}$, $P$ is the precipitation intensity in m a$^{-1}$, $k_w$ is bankfull width per unit scaled discharge in s$^{0.5}$ m$^{-0.5}$, and $S$ is slope (Tucker et al., 2001; Tucker, 2010).



Equating $K\left(\sqrt{A}|\nabla z| - \theta\right)$, i.e., the incision term of Eq. (1), to $D_c$ of Eqs. (A1) we can relate the incision coefficient $K$ and the incision threshold $\theta$ of Eq. (1) with CHILD's parameters according to

$$K = k_b \frac{k_t \sqrt{P}}{k_w} \frac{\sqrt{1\,\text{a}}}{\sqrt{31557600\,\text{s}}} \quad, \tag{A2}$$

$$\tau_c = \frac{k_t \sqrt{P}}{k_w} \theta \frac{\sqrt{1\,\text{a}}}{\sqrt{31557600\,\text{s}}} \quad. \tag{A3}$$

Equations (A2) and (A3) include the unit conversion factor $\sqrt{1\,\text{a}}/\sqrt{31557600\,\text{s}}$ because the input files of CHILD include variables with units of both years and seconds.

In Eqs. (A2) and (A3), we assume constant values of $k_t = 9810\,\text{N m}^{-3}$, $P \approx 1.31\,\text{m a}^{-1}$, and $k_w = 10\,\text{s}^{0.5}\,\text{m}^{-0.5}$, and we obtain the desired values of $K$ and $\theta$ by entering the appropriate values of $k_b$ and $\tau_c$ into CHILD's input files.

**Author contribution**

NT derived analytical results, and NT and JWK interpreted them. NT designed, performed, and analyzed numerical
simulations, and NT and JWK interpreted them. NT drafted the manuscript, and NT and JWK edited it.

**Competing interests**

The authors declare that they have no conflict of interest.

**Acknowledgments**

This study was made possible by financial support from ETH Zurich.

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



**Tables**

Table 1: Values of parameters ($K$, $D$, $U$, and $\theta$) and characteristic scales ($l_c$, $h_c$, and $G_c$) of the landscapes described in Sect. 3.1. All landscapes have equal incision-threshold numbers $N_\theta$ and evolve geometrically similarly. The values of $K$, $D$, and $U$ of the landscapes are less than one order of magnitude smaller or larger than those of Landscape D, which are typical in the literature (e.g., Perron et al., 2008; Tucker, 2009). Values of incision thresholds $\theta$ are such that $N_\theta = K\theta/U = 1$. Maps of landscapes A–D are shown in Figs. 1–3.

| | Simulated landscapes with equal incision-threshold numbers: $N_\theta = K\theta \ / \ U = 1$ | | | | | | |
|---|---|---|---|---|---|---|---|
| Landscape name | Incision coefficient $K$ ($a^{-1}$) | Diffusion coefficient $D$ ($m^2 a^{-1}$) | Uplift rate $U$ ($m a^{-1}$) | Characteristic length $l_c = \sqrt{D/K}$ (m) | Characteristic height $h_c = U/K$ (m) | Characteristic gradient $G_c = h_c/l_c$ ($-$) | Incision threshold $\theta$ (m) |
| A | $10^{-6}$ | $10^{-2}$ | $0.16 \times 10^{-4}$ | 100 | 16 | 0.16 | 16 |
| B | $4 \times 10^{-6}$ | $0.25 \times 10^{-2}$ | $10^{-4}$ | 25 | 25 | 1 | 25 |
| C | $0.5 \times 10^{-6}$ | $2 \times 10^{-2}$ | $0.4 \times 10^{-4}$ | 200 | 80 | 0.4 | 80 |
| D | $10^{-6}$ | $10^{-2}$ | $10^{-4}$ | 100 | 100 | 1 | 100 |
| E | $2 \times 10^{-6}$ | $0.5 \times 10^{-2}$ | $0.4 \times 10^{-4}$ | 50 | 20 | 0.4 | 20 |
| F | $2 \times 10^{-6}$ | $0.5 \times 10^{-2}$ | $2.5 \times 10^{-4}$ | 50 | 125 | 2.5 | 125 |
| G | $0.25 \times 10^{-6}$ | $4 \times 10^{-2}$ | $10^{-4}$ | 400 | 400 | 1 | 400 |
| H | $0.5 \times 10^{-6}$ | $2 \times 10^{-2}$ | $2.5 \times 10^{-4}$ | 200 | 500 | 2.5 | 500 |
| I | $10^{-6}$ | $10^{-2}$ | $6.25 \times 10^{-4}$ | 100 | 625 | 6.25 | 625 |



Table 2: Incision-threshold numbers $N_\theta$ and corresponding incision thresholds $\theta$, parameters $K$, $D$, and $U$, and characteristic scales of the landscapes described in Sect. 3.2. All nine landscapes have the same parameters $K$, $D$, and $U$, and, thus, the same characteristic scales. These landscapes are illustrated in Figs. 4–9.

| Simulated landscapes with different incision-threshold numbers $N_\theta$ | |
| --- | --- |
| Incision-threshold number $N_\theta = K\theta/U$ (−) | Incision threshold $\theta$ (m) |
| 0 | 0 |
| 0.1 | 2.5 |
| 0.2 | 5 |
| 0.4 | 10 |
| 0.8 | 20 |
| 1 | 25 |
| 1.6 | 40 |
| 2 | 50 |
| 4 | 100 |

| Common parameters for all of the above landscapes: | |
| --- | --- |
| Incision coefficient | $K = 2 \times 10^{-6}\ \mathrm{a}^{-1}$ |
| Diffusion coefficient | $D = 0.5 \times 10^{-2}\ \mathrm{m}^2\mathrm{a}^{-1}$ |
| Uplift rate | $U = 0.5 \times 10^{-4}\ \mathrm{ma}^{-1}$ |
| Characteristic length | $l_c = \sqrt{D/K} = 50\ \mathrm{m}$ |
| Characteristic height | $h_c = U/K = 25\ \mathrm{m}$ |
| Characteristic gradient | $G_c = h_c/l_c = 0.5$ |
| Characteristic curvature | $\kappa_c = h_c/l_c^2 = 0.01\ \mathrm{m}^{-1}$ |



Table 3: Lengths, reliefs, and mean slopes of profiles along the longest flow paths of landscapes with different incision-threshold numbers $N_\theta$ (see Table 2 and Sect. 3.2). These profiles are shown in Fig. 9 with their original elevations and in Fig. 10 with their elevations normalized by $N_\theta + 1$. To demonstrate that the dependence of elevations on $N_\theta$ is reversed when profiles are normalized (see Sect. 3.2.2), we show in this table profile elevations at a distance of 0.5 km from the ridge.

| Profiles along the longest flow paths of landscapes of Table 2 | | | | | |
|---|---|---|---|---|---|
| Incision-threshold number, $N_\theta$ | Total length | Total relief | Mean slope | Elevation at 0.5 km away from ridge | |
| | | | | Original, $z$ | Normalized, $z/(N_\theta + 1)$ |
| $(-)$ | (km) | (m) | $(-)$ | (m) | (m) |
| 0 | 4.999 | 653 | 0.13 | 450 | 450 |
| 0.1 | 5.578 | 712 | 0.13 | 485 | 441 |
| 0.2 | 5.033 | 748 | 0.15 | 526 | 439 |
| 0.4 | 5.199 | 898 | 0.17 | 635 | 453 |
| 0.8 | 4.969 | 1090 | 0.22 | 752 | 418 |
| 1 | 5.076 | 1181 | 0.23 | 879 | 439 |
| 1.6 | 5.109 | 1476 | 0.29 | 1052 | 405 |
| 2 | 5.136 | 1581 | 0.31 | 1148 | 383 |
| 4 | 5.053 | 2138 | 0.42 | 1555 | 311 |





## Figures



Figure 1: **Horizontal geometric similarity of landscapes with equal incision-threshold numbers $N_\theta$.** Shaded relief maps show the plan-view geometric similarity of four landscapes with different parameters, but with the same $N_\theta$, and with domains and initial conditions that

5    are normalized by the characteristic scales of length and height $l_c$ and $h_c$. To highlight both that the landscapes are different in size and that they are geometrically similar when normalized by $l_c$, we show domain sizes both in km (top and left, bold fonts) and in units of $l_c$ (bottom and right, normal fonts). Note that the characteristic gradient $G_c$ is not the same across the four landscapes. Thus, the four landscapes have different topographic slopes, which are reflected in the different shades of gray used in the four maps.





Figure 2: **Horizontal and vertical geometric similarity of landscapes with equal incision-threshold numbers $N_\theta$.** Elevation maps show that the four landscapes of Fig. 1 are geometrically similar in the vertical direction as well. We show domain sizes and color-scale elevations both in km or m (top and left, bold fonts) and in units of characteristic length and height $l_c$ and $h_c$ (bottom and right, normal fonts). Note that we use color scales that are normalized by $h_c$, i.e., each color corresponds to the same elevation in units of $h_c$ across all four landscapes. Therefore, the fact that the four maps have the same color pattern shows that their elevations are equivalent when normalized by $h_c$, i.e., the landscapes are geometrically similar.

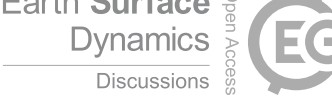

**Figure 3: Horizontal geometric similarity of zones of zero incision.** Red regions show the Voronoi polygons of points with $\sqrt{A}|\nabla z| \leq \theta$, where incision is zero according Eq. (1). The resulting maps show that the zones of zero incision in the four landscapes have geometrically similar horizontal extents.





Figure 4: **Overview of influence of value of incision-threshold number $N_\theta$ on morphology of ridges, hillslopes, and valleys.** Steady-state shaded relief maps show the nine landscapes of Sect. 3.2, which have equal incision coefficients, diffusion coefficients, and uplift rates (i.e., equal characteristic scales), and unequal incision thresholds, such that $N_\theta$ values range from 0 to 4 (see parameters in Table 2 and definition of $N_\theta$ in Eq. 9). The maps are arranged such that $N_\theta$ increases from left to right and from top to bottom. We interpret these shaded relief maps in the description of Fig. 5, where we show enlarged views of a rectangular region from each map to enhance the visibility of landscape features. Here we show these regions with blue rectangles. Their extents are 5 km × 4 km (equivalently, 100 $l_c$ × 80 $l_c$) and are chosen such that they contain each landscape's longest flow path and the corresponding drainage basin. We mark these flow paths with blue lines, and we present profile plots along their course in Figs. 9 and 10.





Figure 5: **Influence of incision-threshold number $N_\theta$ on morphology of ridges, hillslopes, and valleys.** Shaded relief plots corresponding to the blue rectangular regions in Fig. 4 are arranged such that $N_\theta$ increases from left to right and from top to bottom (identical to Fig. 4; see parameter values in Table 2). The illumination angle is consistent among all panels; thus greater contrasts in gray scale correspond to steeper slopes. Maps with higher $N_\theta$ have steeper slopes, as indicated by the greater contrast. Maps with higher $N_\theta$ also exhibit wider ridges and interfluves (note the distance between tips of valley networks and basin or sub-basin divides), with the result that ridges and hillslopes appear smoother in plan view.




Earth **Surface**
Dynamics
Discussions


Figure 6: **Increase of relief as the incision-threshold number $N_\theta$ increases.** Steady-state elevation maps of the nine landscapes of Sect. 3.2 (parameter values in Table 2) are plotted using a single elevation color scale, facilitating visual comparison of elevations across landscapes. The blue lines show the longest flow path of each landscape and the blue rectangles mark the regions that are magnified in Figs. 5, 7, and 8. The landscapes are arranged such that $N_\theta$ increases from left to right and from top to bottom. By comparing the colors of the maps, we observe that landscapes with higher $N_\theta$ values have greater relief (see also the maximum elevation of each landscape, displayed at the bottom of each map).



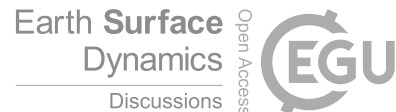

Figure 7: **Expanding extent of zones of zero incision as the incision-threshold number $N_\theta$ increases.** Red regions indicate Voronoi polygons of points with $\sqrt{A}|\nabla z| \leq \theta$, where incision does not operate (Eq. 1), and white indicates the remaining areas where incision operates. Note that the landscape with $N_\theta = 0$ follows Eq. (2) which is not defined piecewise; thus, zones of zero incision are not defined
5  for this landscape. As $N_\theta$ increases, the zones of zero incision become more extensive, and eventually occupy almost all ridges and hillslopes. In the maps of the top row, which have the smallest of the examined $N_\theta$ values, zero-incision zones appear mainly along divides of major drainage basins. In the maps of the middle row, which have moderate $N_\theta$ values that do not exceed 1, zero-incision zones completely cover the main drainage divides and increasingly appear on smaller divides (interfluves) and on hillslopes. In the first two maps of the bottom row, which have $N_\theta$ equal to 1.6 and 2, zero-incision zones occupy increasingly large portions of hillslopes, and in the
10  third map of the bottom row, which has $N_\theta = 4$, they almost completely cover the hillslopes, with the white areas following the dendritic patterns of the valley network, which can be seen also in Fig. 8. Thus, for the largest of the examined $N_\theta$ values, incision operates almost exclusively in valleys and is largely non-existent on the hillslopes.

Earth **Surface**
**Dynamics**
Discussions
EGU



**Figure 8: Deeper and sparser valleys, and wider hillslopes, in landscapes with higher incision-threshold numbers $N_\theta$.** Steady-state maps of the Laplacian curvature $\nabla^2 z$ of the landscapes of Sect. 3.2 reveal how valley networks and hillslopes change as $N_\theta$ increases. Areas with $\nabla^2 z \leq 0$ are shown in white and areas with $\nabla^2 z > 0$ are shown in grayscale. Gray dendritic patterns indicate valley networks, because concave areas can be considered as valleys, and convex areas as ridges or hillslopes (e.g., Howard, 1994). As $N_\theta$ increases, ridges and hillslopes become wider, and gray dendritic valley patterns become sparser. The color scales of the nine maps are not the same; as $N_\theta$ increases, the maximum value of curvature increases and, thus, curvature has a wider range of positive values. Therefore, as $N_\theta$ increases, concave areas become more concave, i.e., valleys become deeper. By contrast, the minimum value of curvature is $\nabla^2 z = -\kappa_c$ in all color scales and, thus, the most convex areas are equally convex in all maps. However, the extent of these most convex areas becomes wider as $N_\theta$ increases, because the value $\nabla^2 z = -\kappa_c$ corresponds to zones of zero incision (see Sect. 2.2), which become wider as $N_\theta$ increases (see Fig. 7). Therefore, as $N_\theta$ increases, hillslopes become more convex because bigger portions of them have the minimum value of curvature.



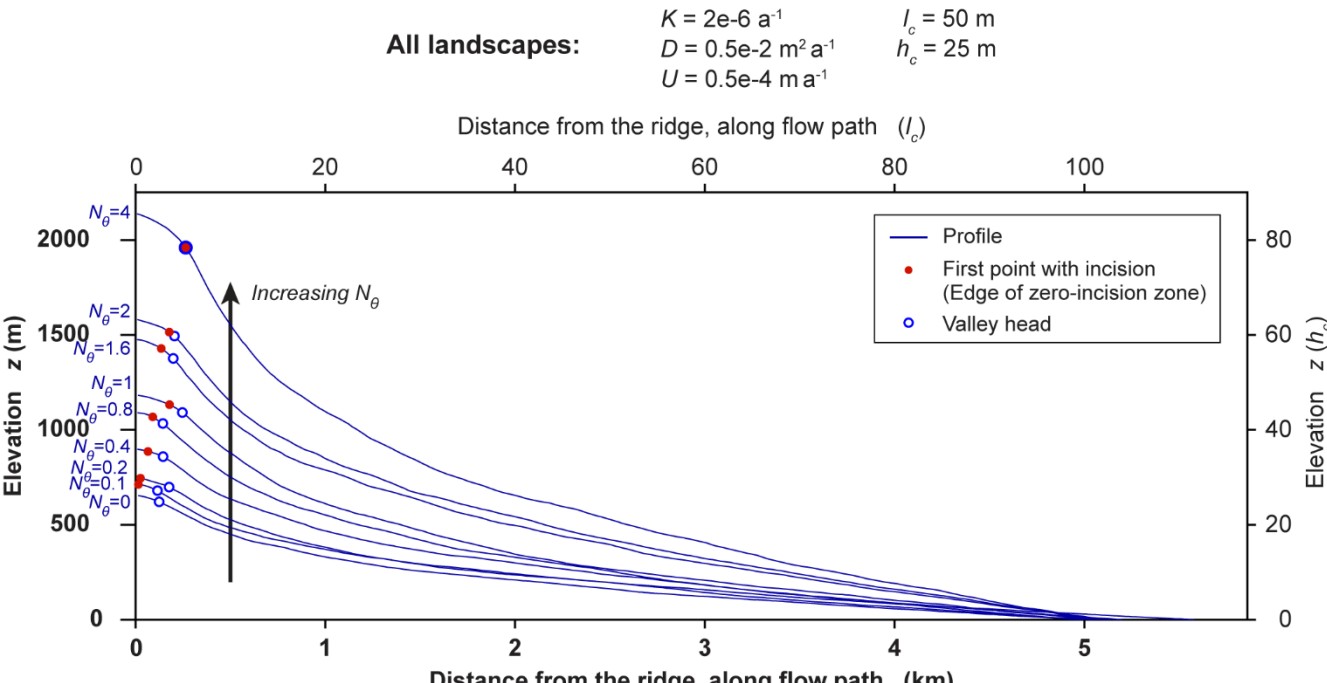

Figure 9: **Steepening of profiles as the incision-threshold number $N_\theta$ increases.** Blue profiles show elevation $z$ versus distance from the ridge (along the flow path) for the longest flow paths in each of the nine landscapes of Sect. 3.2. These flow paths are marked with blue lines on the maps of Figs. 4–8. All profiles have roughly equal horizontal lengths of ~5 km, or ~100 $l_c$ (see lengths in Table 3). As $N_\theta$ increases, the total reliefs of profiles (i.e., their elevations at the ridge) increase and, thus, their slopes become steeper (see reliefs and mean slopes in Table 3). On each profile, a red dot shows the edge of the zero-incision zone, defined here as the first point along the profile with $\sqrt{A}|\nabla z| > \theta$, i.e., the first point with incision, and a blue circle shows the first-order valley head, defined as the first point with non-negative curvature ($\nabla^2 z \geq 0$). (We do not show a red dot for $N_\theta = 0$, for which zero-incision zones do not exist.) As $N_\theta$ increases, the red dots and the blue circles tend to move away from the ridge, indicating that the zero-incision zones become wider and the drainage density decreases as $N_\theta$ increases. Note that the edges of the zero-incision zones are more sensitive to $N_\theta$ than the valley heads are. Thus, as $N_\theta$ increases, the red dots and blue circles converge, becoming indistinguishable for $N_\theta = 4$.



Earth **Surface**
**Dynamics**
Discussions



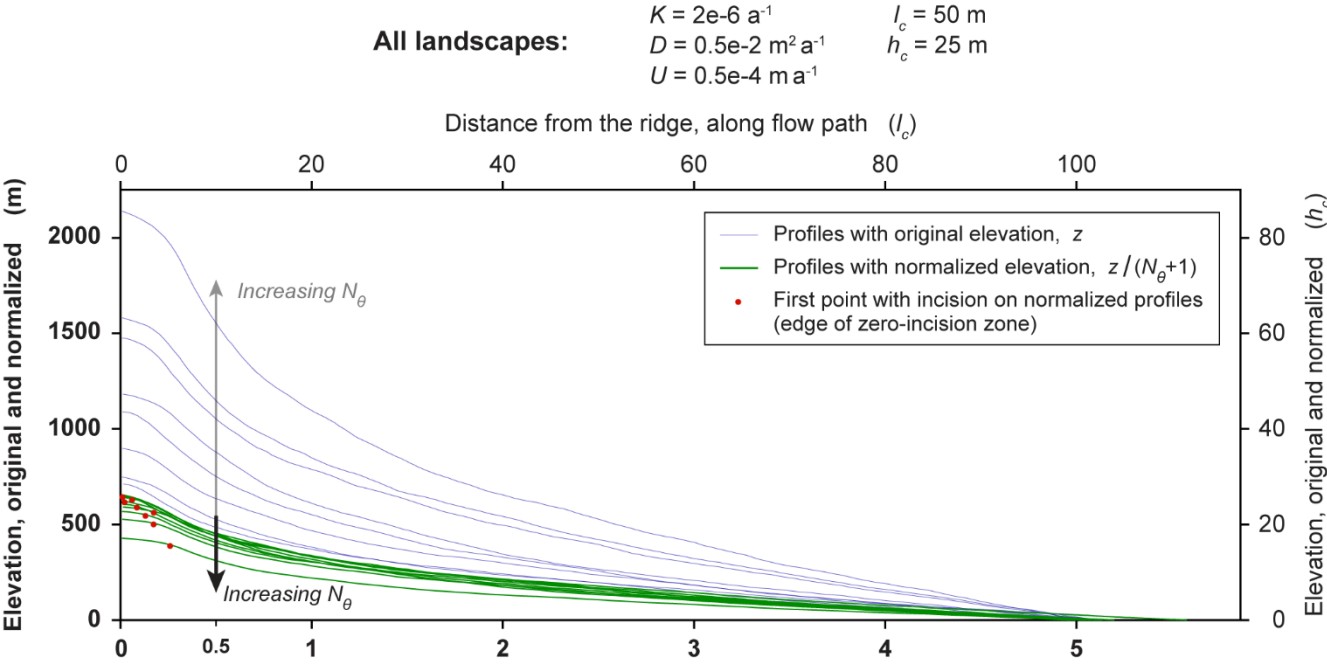

Figure 10: **Equivalence of elevations that are normalized by $N_\theta + 1$.** Green lines show the profiles of Fig. 9 (shown again here with light blue lines), normalized by dividing by $N_\theta + 1$. The normalized profiles largely collapse on each other. Along each profile, this tendency becomes stronger in the downstream direction, where the distance from the zone of zero incision grows (the edges of zero-incision zones are indicated by red dots). As $N_\theta$ increases, the normalized profile elevations generally decrease slightly, whereas the original profile elevations increase substantially (see Table 3, which gives elevations, original and normalized, at a distance of 0.5 km from the ridge, which is roughly the location of the black arrow in this figure).