# Peer review of "Dimensional analysis of a landscape evolution model with incision threshold"

_Earth Surface Dynamics, 2019_

## Referee Comment (RC1) · Eric Deal (Referee) · 12 Feb 2020

In "Dimensional analysis of a landscape evolution model with incision threshold", the authors extend a previous analysis of the classic advection-diffusion landscape evolution equation with a constant source term to include a threshold for erosion. Though the neither the idea of erosion thresholds nor the advection-diffusion equation themselves are new, I have not seen them combined in this way before. This novel connection, together with the authors very thoughtful and insightful analysis of the equations produces some valuable and widely applicable conclusions. In particular, the fact that introducing a threshold adds a parameter to the nondimensionalized equations that

distinguishes landscapes with different relative threshold magnitudes from one another fundamentally is very interesting.

Overall I find the paper to be well cited, novel, scientifically rigorous and the impact is appropriate for the journal. The authors are clearly knowledgeable of the state of the art, and have placed their work in the correct context. The writing, figures and overall presentation is excellent. One of my few criticisms is that the authors have a tendency to over-explain some concepts, and I think it would be possible to shorten some explanations and derivations. However, the paper is not too long, and I don't think that this is a necessary change.

I have one significant criticism, which is that the authors have used a threshold with a steady-state constant rainfall/discharge, yet have compared it in many ways to thresholds which are derived under the assumption of stochastic forcing. There is very little modern work on erosion thresholds outside of a stochastic forcing context, because without a stochastic forcing, thresholds lead to dramatic, very nonlinear behaviours which are not realistic. I find that it is not difficult to include a simple stochastic forcing, though it would require rerunning the models shown in the paper. I think that the effort required to use stochastic forcing would be rewarded with a much firmer theoretical connection to modern work on incision thresholds and more interest from the community. I have included a document which contains further argument for using a stochastic forcing.

Besides this one major criticism, I would enthusiastically recommend this paper for publication. The authors do include a small section addressing my criticism already, and the novelty of the approach still stands even if the authors do not adopt stochastic forcing. I will not withhold my recommendation for publication contingent on addressing this point. However, I do want to take the opportunity to stress that I feel that including stochastic forcing will increase the significance of the paper, increase how well it fits in with the state of the art, as well as increase its impact, and I very strongly urge the authors to consider redoing the analysis with stochastic forcing, or adding it alongside

the original analysis.

- Eric Deal

Please also note the supplement to this comment:
https://www.earth-surf-dynam-discuss.net/esurf-2019-80/esurf-2019-80-RC1-supplement.pdf
* * *
[Figure]

**Supplement:**

**Stochastic threshold incision**

February 12, 2020
Eric Deal

**Motivation**

Modern work on fluvial incision thresholds started with Tucker and Bras [2000], and since then all the major work on fluvial erosion thresholds has included a stochastic treatment [Wolman and Miller, 1960, Tucker and Bras, 2000, Snyder et al., 2003, Tucker, 2004, Lague et al., 2005, DiBiase and Whipple, 2011, Lague, 2014, Rossi et al., 2016, Scherler et al., 2017, Deal et al., 2018]. As Lague [2014] points out, if you try to use a threshold with constant forcing, you have major nonlinearities in steady-state slope, critical discharge, critical area, etc, when the magnitude of the forcing is close the the magnitude of the threshold. There are likely few rivers in the world whose discharge is anything steady enough to be considered remotely constant, and at the same time, it is likely that many rivers have a threshold of motion that is not exceeded at least part of the time. Truly, the concepts of stochastic forcing and fluvial erosion threshold different aspects of the same process.

**Applying a stochastic forcing**

Start with

$$\frac{\partial z}{\partial t} = U - D\Delta z + E \tag{1}$$

where $E$ is the fluvial incision. To properly do the incision with a threshold, it is critical to take into account the variability in flow. This allows for incision to occur "below" the threshold, so there is not a singularity when $K\sqrt{A}|\nabla z| = \theta$. We can do this by considering a short timescale incision model

$$I = K\sqrt{A}|\nabla z|q^* - K\theta \tag{2}$$

where $q^*$ is the discharge normalized by the mean. The simplest option would be to consider that $q^*$ is exponentially distributed with a mean of 1,

$$p(q^*) = e^{-q^*} \tag{3}$$

Then the fluvial incision rate is

$$E = \int_{q_c^*}^{\infty} I(q^*)p(q^*)dq^* = K\sqrt{A}|\nabla z| \int_{q_c^*}^{\infty} (q^* - q_c^*)\, e^{-q^*} dq^* \tag{4}$$

where $q_c^*$ is found by solving $I(0)$, leading to $q_c^* = \frac{\theta}{\sqrt{A}|\nabla z|}$. The solution to the integral is simply

$$E = K\sqrt{A}|\nabla z| e^{-\frac{\theta}{\sqrt{A}|\nabla z|}} \tag{5}$$

Therefore, the full PDE is

$$\frac{\partial z}{\partial t} = U - D\Delta z + K\sqrt{A}|\nabla z| e^{-\frac{\theta}{\sqrt{A}|\nabla z|}} \tag{6}$$

If we nondimensionalize in the same way as in the paper, this leads to

$$\frac{\partial z^*}{\partial t^*} = 1 - \Delta^* z^* + k_s^* e^{-N_\theta/k_s^*} \tag{7}$$

where $k_s^* = \sqrt{A^*}|\nabla^* z^*|$ and all other parameters are exactly as in the paper. Note, now the PDE is no longer piecewise, but instead there is an exponential decrease to zero in the magnitude of the fluvial incision term in the neighbourhood of $k_s^* \approx N_\theta$.

**Possible interpretations of $N_\theta$**

You mention that there are 3 characteristic scales in the problem. I think that $\theta$ implies a fourth scale, which is fundamentally the characteristic grain size in the landscape. The characteristic grain size $D_c$ sets the threshold of motion. In the simplest case, it will be linearly proportional to the threshold itself [e.g. Scherler et al., 2017], so we can say $D_c \propto \theta$. Many papers on threshold stochastic formulations point out that the ratio of $\frac{\Psi}{U} = \frac{K\theta}{U}$ is the key nondimensional number that describes the importance of thresholds [e.g. DiBiase and Whipple, 2011, Lague, 2014].

I also point out in Deal et al. [2018] that this ratio is similar, and when everything is linear, identical to the ratio of the discharge that can move the mean grain size to the mean discharge $\frac{q_c}{\mu}$. Since the discharge and area are the same here, the mean discharge (cast as a flow depth) is $\sqrt{A}$, and the critical discharge is the flow depth that can move the mean grain size for a given slope, here we take the characteristic slope $\theta = q_c G_c$, so the critical discharge is $q_c = \frac{\theta}{G_c}$. The same result can be found by setting $I(q^*) = 0$ and solving for the critical area at the mean discharge $q^* = 1$ and the characteristic slope $G_c$. Then we see that if we look at the nondimensional critical discharge at the characteristic area and slope

$$\frac{q_c}{\mu_c} = \frac{q_c}{\sqrt{A_c}} = \frac{q_c}{l_c} = \frac{\theta}{l_c G_c} = \frac{K\theta}{U} = N_\theta. \tag{8}$$

So $N_\theta$ can be thought of as referring directly to the channel hydraulics at the characteristic area. In fact there are many perspectives on it: $N_\theta \propto \frac{D_c}{h_c}$ can be thought of as a nondimensional grain size, scaled by $h_c$; or as a nondimensional critical discharge/flow depth $N_\theta = \frac{q_c}{l_c}$, scaled by $l_c$; or as a nondimensional critical area to reach the critical discharge at mean flow $N_\theta = \frac{A_\theta}{A_c}$, $(A_\theta = q_c^2)$; or as a nondimensional shear stress $N_\theta = \frac{\theta}{\theta_c}$, scaled by the characteristic shear stress $\theta_c = \sqrt{A_c} G_c$.

One of the main conclusions of earlier studies is that whether grains are considered large or small is a function of the uplift rate. When uplift rates are high, grains are relatively smaller. This is captured very nicely by $N_\theta$.

**References**

E. Deal, J. Braun, and G. Botter. Understanding the Role of Rainfall and Hydrology in Determining Fluvial Erosion Efficiency. *Journal of Geophysical Research: Earth Surface*, 123(4):744–778, 2018. ISSN 2169-9003. doi: 10.1002/2017jf004393.

R. A. DiBiase and K. X. Whipple. The influence of erosion thresholds and runoff variability on the relationships among topography, climate, and erosion rate. *Journal of Geophysical Research: Earth Surface (2003-2012)*, 116(F4), 2011. doi: 10.1029/2011JF002095.

D. Lague. The stream power river incision model: evidence, theory and beyond. *Earth Surface Processes and Landforms*, 39(1):38–61, 2014. doi: 10.1002/esp.3462.

D. Lague, N. Hovius, and P. Davy. Discharge, discharge variability, and the bedrock channel profile. *Journal of Geophysical Research: Earth Surface (2003-2012)*, 110(F4), 2005. doi: 10.1029/2004JF000259.

M. W. Rossi, K. X. Whipple, and E. R. Vivoni. Precipitation and evapotranspiration controls on daily runoff variability in the contiguous United States and Puerto Rico. *Journal of Geophysical Research: Earth Surface*, 2016. doi: 10.1002/2015JF003446.

D. Scherler, R. A. DiBiase, G. B. Fisher, and J. Avouac. Testing monsoonal controls on bedrock river incision in the Himalaya and Eastern Tibet with a stochastic‐threshold stream power model. *Journal of Geophysical Research: Earth Surface*, 122(7):1389–1429, 2017. ISSN 2169-9011. doi: 10.1002/2016jf004011.

N. P. Snyder, K. X. Whipple, G. E. Tucker, and D. J. Merritts. Importance of a stochastic distribution of floods and erosion thresholds in the bedrock river incision problem. *Journal of Geophysical Research: Solid Earth (1978-2012)*, 108(B2), 2003. doi: 10.1029/2001JB001655.

G. E. Tucker. Drainage basin sensitivity to tectonic and climatic forcing: implications of a stochastic model for the role of entrainment and erosion thresholds. *Earth Surface Processes and Landforms*, 29(2):185–205, 2004. doi: 10.1002/esp.1020.

G. E. Tucker and R. L. Bras. A stochastic approach to modeling the role of rainfall variability in drainage basin evolution. *Water Resources Research*, 36(7):1953–1964, 2000.

G. M. Wolman and J. P. Miller. Magnitude and frequency of forces in geomorphic processes. *The Journal of Geology*, pages 54–74, 1960.

---

## Referee Comment (RC2) · Anonymous Referee #2 · 12 Feb 2020

The authors present a dimensional analysis on a landscape evolution model that includes an incision threshold. In their analysis, they non-dimensionalize a landscape evolution model with an incision threshold using the same length, height, and time scales from their previous analysis (Theodoratos et al., 2018). They show that adding a threshold changes the model from a 0-parameter model to 1-parameter model, named $N\theta$. In this model, simulations using the same value of $N\theta$ show geometric similarity, and simulations increasing $N\theta$ cause an increase in dimensionless relief with $N\theta$ + 1. Last, they find increasing $N\theta$ effectively causes fluvial processes to be isolated to valleys.

[Figure]

This manuscript is a thorough and straightforward presentation of a landscape evolution model including an incision threshold. This analysis is a logical and sensible next-step after the authors' previous work (Theodoratos et al., 2018), yet their results and discussion are still novel. The manuscript is well-written and understandable. I recommend acceptance of this manuscript without any modification. I look forward to the future work of the authors.

Line Comments: Page 2, Line 9-12: Interesting thought. Is there a way to implement an intermittency factor into the model to simulate this?

Page 3, Line 16: How do you expect your results of varying N$\theta$ to change with a non-linear hillslope diffusion formulation?

Page 3, Line 29 to Page 4, Line 8: Could this paragraph by summarized into a table or set of equations?

Page 5, Line 7-8: What sets the maximum value of curvature in Figure 8?

Page 7, Line 32-38: What is the rationale behind scaling the initial conditions with the length and height scales? Without this that the landscape will not be geometrically similar, but I'm not sure why the height of the initial randomization and topography should scale with parameters. A super bumpy initial surface may look flat at a zoomed-out scale and vice versa.

Page 8, Line 28-34: Great illustrations of how your dimensional analysis works.

Page 9: A good set of illustrations to show the effects of varying N$\theta$. Through explanation.

Page 11, Line 1-2: I find this discussion though provoking. What sets the value of N$\theta$ = 4? and how would your results change if the initial condition was more or less bumpy?

Page 12, Line 27-29: Can the authors hypothesize what this additional effect is? Are the smaller catchments becoming more diffusional as the incision threshold is increased? This would increase the positive curvature in the valleys, but perhaps this increase in curvature does not scale with $1+ N\theta$. Could plotting how the ratio between area of no incision and the total area with $N\theta$ be informative?

---

## Referee Comment (RC3) · Wolfgang Schwanghart (Referee) · 4 Mar 2020

Nikos Theodoratos and James Kirchner conduct a dimensional analysis of the stream power incision model that includes an incision threshold that defines zones of zero incision below a defined stream power threshold. The dimensional analysis reveals that the incision threshold number remains the only parameter that governs the evolution of landscapes simulated by the stream power incision model. Their analysis expands on and complements a previous paper, that the authors also published in ESURF.

Overall, the paper is very well written and illustrated. I like the comprehensive explanation which allows readers to follow the steps taken by the authors. The results are well

illustrated by the figures and tables, but the text could be abbreviated. The discussion places the results in the context of stochastic stream power incision models, and the choice of characteristic scales. In my opinion, the discussion could also revisit some of the assumption behind the study (in particular uniform uplift).

All in all, I think that the paper is already in a very good shape and ready to be published in ESURF after minor revisions.

Specific comments:

7-17 While referring the reader to Theodoratos et al. (2018), you may nevertheless provide some more details on the numeric simulations here, e.g. resolution or the nr of vertices used in the TIN. 13-19 Given Eq. 1, this should rather read that points with any given stream power above the threshold value experience a stream power greater than zero, ...

---

## Author Comment (AC1) · 4 Mar 2020

**Reply to referee comments on "Dimensional analysis of a landscape evolution model with incision threshold" by Theodoratos and Kirchner**

We are grateful to Eric Deal, Wolfgang Schwanghart, and the anonymous referee for their feedback on our manuscript. In the following response to their reviews, we first quote their comments (in blocks of italic text) and then respond (in normal text).

Nikos Theodoratos and James Kirchner

**1. Response to referee Eric Deal**

*In "Dimensional analysis of a landscape evolution model with incision threshold", the authors extend a previous analysis of the classic advection-diffusion landscape evolution equation with a constant source term to include a threshold for erosion. Though the neither the idea of erosion thresholds nor the advection-diffusion equation themselves are new, I have not seen them combined in this way before. This novel connection, together with the authors very thoughtful and insightful analysis of the equations produces some valuable and widely applicable conclusions. In particular, the fact that introducing a threshold adds a parameter to the nondimensionalized equations that distinguishes landscapes with different relative threshold magnitudes from one another fundamentally is very interesting.*

*Overall I find the paper to be well cited, novel, scientifically rigorous and the impact is appropriate for the journal. The authors are clearly knowledgeable of the state of the art, and have placed their work in the correct context. The writing, figures and overall presentation is excellent.*

Thank you.

*One of my few criticisms is that the authors have a tendency to over-explain some concepts, and I think it would be possible to shorten some explanations and derivations. However, the paper is not too long, and I don't think that this is a necessary change.*

As you can see below, the anonymous referee recommends to summarize in a table the dimensions of variables and parameters of the governing equation (now presented as a paragraph in pages 3 and 4). We like this idea, and we will apply it to other parts of the paper, such as the derivations of characteristic scales and of various dimensionless terms (pages 4 and 5). This will shorten the derivations.

*I have one significant criticism, which is that the authors have used a threshold with a steady-state constant rainfall/discharge, yet have compared it in many ways to thresholds which are derived under the assumption of stochastic forcing. There is very little modern work on erosion thresholds outside of a stochastic forcing context, because without a stochastic forcing, thresholds lead to dramatic, very nonlinear behaviours which are not realistic. I find that it is not difficult to include a simple stochastic forcing, though it would require rerunning the models shown in the paper. I think that the effort required to use stochastic forcing would be rewarded with a much firmer theoretical connection to*

*modern work on incision thresholds and more interest from the community. I have included a document which contains further argument for using a stochastic forcing.*

*Besides this one major criticism, I would enthusiastically recommend this paper for publication. The authors do include a small section addressing my criticism already, and the novelty of the approach still stands even if the authors do not adopt stochastic forcing. I will not withhold my recommendation for publication contingent on addressing this point. However, I do want to take the opportunity to stress that I feel that including stochastic forcing will increase the significance of the paper, increase how well it fits in with the state of the art, as well as increase its impact, and I very strongly urge the authors to consider redoing the analysis with stochastic forcing, or adding it alongside the original analysis.*

We thank you for recommending our paper for publication. We acknowledge your comment and we welcome your suggestions. We thank you for elaborating your suggestions in a supplement.

In addition to the problems that you describe, the incision threshold formulation that we adopt here "has the disadvantage of being unable to explore the non-linear relationship between average precipitation and long-term incision rates that we describe in the Introduction (Sect. 1)", as we mention in the manuscript. We agree that these effects of erosion thresholds can be much better understood within the context of stochastic precipitation.

In this work, however, we do not examine the effects of thresholds on long-term incision rates. Rather, we examine the effects of thresholds on the geometric similarity that we found in Theodoratos et al. (2018). Specifically, we examine whether we can use $l_c$, $h_c$, and $t_c$ to obtain geometrically similar landscapes if these landscapes include incision thresholds. We find that landscapes can be geometrically similar in the specific case of having equal incision-threshold numbers $N_\theta$. Further, we examine how dissimilar they are when their $N_\theta$ are not equal, and we find that this depends on the value of $N_\theta$.

Reformulating the precipitation from constant to stochastic could affect the definition of the incision coefficient $K$, e.g., see Eqs. (21) and (22) in Tucker (2004). This would, in turn, affect the definitions of characteristic scales (Eqs. 3–5). In this way, the effects that we examine here could be obscured. Therefore, we prefer to examine one change at a time, by examining the inclusion of a threshold in this paper, and examining stochastic precipitation in a new manuscript.

Making the precipitation stochastic will likely lead to significant changes to the results and interpretations of the current manuscript. Therefore, a separate manuscript would be a more suitable way to present changes. Furthermore, with your supplement, you have made a substantial contribution toward new results and interpretations. Therefore, we hope you would accept an invitation to be a co-author of the future manuscript.

**2. Response to the anonymous referee**

*The authors present a dimensional analysis on a landscape evolution model that includes an incision threshold. In their analysis, they non-dimensionalize a landscape evolution model with an incision threshold using the same length, height, and time scales from their previous analysis (Theodoratos et al., 2018). They show that adding a threshold changes the model from a 0-parameter model to 1-parameter model, named $N_\theta$. In this model, simulations using the same value of $N_\theta$ show geometric similarity, and simulations*

*increasing $N_\theta$ cause an increase in dimensionless relief with $N_\theta + 1$. Last, they find increasing $N_\theta$ effectively causes fluvial processes to be isolated to valleys.*

*This manuscript is a thorough and straightforward presentation of a landscape evolution model including an incision threshold. This analysis is a logical and sensible next-step after the authors' previous work (Theodoratos et al., 2018), yet their results and discussion are still novel. The manuscript is well-written and understandable. I recommend acceptance of this manuscript without any modification. I look forward to the future work of the authors.*

Thank you for the supportive comments.

*Line Comments:*

*Page 2, Line 9-12: Interesting thought. Is there a way to implement an intermittency factor into the model to simulate this?*

First of all, the ideas discussed in these lines are not ours, so credit should go, for example, to DiBiase and Whipple (2011), as cited in the manuscript. Intermittency could be simulated, for example, by using stochastic precipitation (see also our response to Eric Deal, above).

*Page 3, Line 16: How do you expect your results of varying $N\theta$ to change with a nonlinear hillslope diffusion formulation?*

We address this question further below, after providing some background information in the answer to a different question.

*Page 3, Line 29 to Page 4, Line 8: Could this paragraph by summarized into a table or set of equations?*

Thank you, this is a nice idea. We will adopt it and extend it. Specifically, we will also summarize in tables the content of:
- lines 4–8, 10, and 16–17 in page 3, where we describe the terms of the governing equation (Eq. 1)
- line 21 in page 4 until line 19 in page 4, where we define the characteristic scales $A_c$, $G_c$, and $\kappa_c$, and we non-dimensionalize terms of the governing equation.

*Page 5, Line 7-8: What sets the maximum value of curvature in Figure 8?*

For a given combination of characteristic curvature $\kappa_c$ and incision-threshold number $N_\theta$, the maximum curvature is mainly controlled by the size of the domain. Specifically, a larger domain leads to larger maximum curvature value.

*Page 7, Line 32-38: What is the rationale behind scaling the initial conditions with the length and height scales? Without this that the landscape will not be geometrically similar, but I'm not sure why the height of the initial randomization and topography should scale with parameters. A super bumpy initial surface may look flat at a zoomed-out scale and vice versa.*

We explain this in more detail in Appendix B of Theodoratos et al. (2018). In brief, two landscapes can be geometrically similar at some time step if they were geometrically similar at the previous time step. Thus, to reach geometrically similar steady states, they must start from geometrically similar initial conditions.

Note that evolving landscapes must be compared at times that are normalized by each landscape's characteristic timescale. For example, if a landscape has characteristic timescale $t_c$ and another landscape has $2t_c$, then a snapshot of the first landscape with some age $t_0$ must be compared with a snapshot of the second landscape with age $2t_0$.

It is true that the bumpiness of surfaces would change when looked at different zoom levels, but this is desirable. We treat lengths and heights as dimensionally distinct quantities and we normalize them independently according to the characteristic length and height $l_c$ and $h_c$ of each landscape. In this way, we rescale drainage areas, slopes, and curvatures such that they result in incision and diffusion rates with appropriate proportions relative to the uplift rates. In the four shaded relief maps of Fig. 1, the differences in shading intensity illustrate how the landscapes' steady states differ in bumpiness. Their initial conditions differed in bumpiness in the same manner.

> *Page 8, Line 28-34: Great illustrations of how your dimensional analysis works.*

Thank you!

> *Page 9: A good set of illustrations to show the effects of varying Nθ. Through explanation.*

Thank you.

> *Page 11, Line 1-2: I find this discussion though provoking. What sets the value of Nθ=4? and how would your results change if the initial condition was more or less bumpy?*

To rephrase your question, what controls the distance between the edge of zones of zero incision and hillslope–valley transitions (e.g., between red dots and blue circles in Fig. 9)?

Below, we qualitatively describe how the topology of flow paths influences this distance. A quantitative prediction of this distance can likely be obtained only by running simulations and analyzing their results.

In what follows, we refer to the edges of zero incision zones as "zone edges", and to hillslope–valley transitions as "valley heads".

To begin with, if the distance between zone edges and valley heads is roughly equal to or smaller than the resolution, then these points will coincide, or be immediate neighbors. So in the rest of this answer, we assume a fixed resolution.

According to the governing equation (Eq. 1), zone edges are points with $\sqrt{A}|\nabla z| = \theta$. Valley heads can be defined as points with Laplacian curvature $\nabla^2 z = 0$, where the topography transitions from being convex to being concave (e.g., Howard, 1994). Setting $\nabla^2 z = 0$ and $\partial z/\partial t = 0$ (steady state), and given the definition of the characteristic height $h_c$ (Eq. 4), the governing equation gives the value $\sqrt{A}|\nabla z| = h_c + \theta$ for valley heads. Finally, given the definition of the incision-threshold number $N_\theta$ (Eq. 9), we see that at zone heads $\sqrt{A}|\nabla z| = N_\theta h_c$ and at valley heads $\sqrt{A}|\nabla z| = (N_\theta + 1)h_c$. Therefore, in steady state, the distance between zone edges and valley heads is equal to the distance needed for the quantity $\sqrt{A}|\nabla z|$ to grow from $N_\theta h_c$ to $(N_\theta + 1)h_c$.

First, we examine how the growth of $\sqrt{A}|\nabla z|$ depends on the convergence or divergence of flow paths. Afterwards, we examine how the convergence or divergence of flow paths is related to the value of $N_\theta$.

Moving downhill along a flow path, the rate of growth of $\sqrt{A}|\nabla z|$ per unit length depends on the rate of drainage area accumulation and on the rate of slope change. (Note that drainage area can only grow, but slope can become either steeper or gentler as we move downstream. So for $\sqrt{A}|\nabla z|$ to grow, drainage area must be accumulated fast enough to compensate for a decreasing slope.)

The rate of drainage area accumulation per unit length depends on the contour curvature, which expresses the convergence or divergence of flow paths. The rate of change of slope per unit length depends on the profile curvature. For definitions of these curvatures, see Mitasova and Hofierka (1993). It can be shown that the Laplacian curvature $\nabla^2 z$ can be expressed as a weighted sum of contour and profile curvature. Thus, for a given value of $\nabla^2 z$, the values of contour and profile curvature are interrelated, specifically, if one of them is increased, then the other must be decreased.

Where flow paths converge more strongly, drainage area accumulates faster. Moreover, contour curvature is larger and thus profile curvature must be more negative (so that a negative Laplacian curvature is obtained, since the point in question must have a curvature between 0 and $-\kappa_c$, the value in zones of zero incision). Thus, slope also becomes steeper faster. Consequently, the quantity $\sqrt{A}|\nabla z|$ has the fastest growth rate per unit length at areas with strongly convergent flow paths.

In landscapes with smaller $N_\theta$, the zone edge is closer to the ridgeline, but in landscapes with larger $N_\theta$, the zone edge is farther from the ridge. Close to the ridge, flow paths are divergent or only weakly convergent. By contrast, farther from the ridge, the topography can become convergent even on hillslopes, because the landscape surface is adjusting itself to the valley network downhill. Therefore, as $N_\theta$ increases, the zone edge moves farther from the ridge to areas where flow paths are more convergent, i.e., to areas where the quantity $\sqrt{A}|\nabla z|$ grows faster from $N_\theta h_c$ to $(N_\theta + 1)h_c$.

To summarize, larger $N_\theta$ results in zone edges that are farther from the ridge, which results in more strongly convergent flow paths, which in turn result in smaller distance between zone edges and valley heads. However, the exact value of $N_\theta$ that leads to a distance smaller than some limit (e.g., smaller than the resolution) may be quantifiable only by simulations.

Beyond the resolution, it seems likely that this value of $N_\theta$ is ultimately influenced by the size and shape of the domain, which influence the large-scale geometry and topology of the valley network, which in turn influence the way that hillslopes are folded to smoothly connect valleys and ridges.

Consequently, the bumpiness of initial conditions may affect the results if it can affect the convergence or divergence of the final landscape surface.

> *Page 3, Line 16: How do you expect your results of varying Nθ to change with a nonlinear hillslope diffusion formulation?*

Nonlinear hillslope diffusion affects the resulting topography of hillslopes, e.g., the way that topographic gradients change along flow paths, or the convergence and divergence of flow paths. Therefore, nonlinear diffusion could change the arrangements of zero incision zones, etc.

> *Page 12, Line 27-29: Can the authors hypothesize what this additional effect is? Are the smaller catchments becoming more diffusional as the incision threshold is increased? This would increase the positive curvature in the valleys, but perhaps this increase in curvature does not scale with 1+ Nθ. Could plotting how the ratio between area of no incision and the total area with Nθ be informative?*

Based on the above discussion regarding flow path convergence and the per-unit-length growth of the quantity $\sqrt{A}|\nabla z|$, we hypothesize that the value of $N_\theta$ affects the structure, as it were, of the landscape. Landscapes with different $N_\theta$ values are composed of areas with zero, weak, medium, and strong incision that are mixed in different proportions. These differences not only result in different incision rates and reliefs, but also result in different competition between incision and diffusion, which controls the horizontal scales of landscapes. Furthermore, the differences in horizontal scales are not constant across landscapes, but instead they are more pronounced in hillslopes and less pronounced in valleys, where the quantity $\sqrt{A}|\nabla z|$ is large relative to the incision threshold $\theta$. We examine effects like these in a separate manuscript that we will submit soon.

**3. Response to referee Wolfgang Schwanghart**

*Nikos Theodoratos and James Kirchner conduct a dimensional analysis of the stream power incision model that includes an incision threshold that defines zones of zero incision below a defined stream power threshold. The dimensional analysis reveals that the incision threshold number remains the only parameter that governs the evolution of landscapes simulated by the stream power incision model. Their analysis expands on and complements a previous paper, that the authors also published in ESURF.*

*Overall, the paper is very well written and illustrated. I like the comprehensive explanation which allows readers to follow the steps taken by the authors. The results are well illustrated by the figures and tables, but the text could be abbreviated.*

Thank you. We will make some abbreviations of the text, as described above.

*The discussion places the results in the context of stochastic stream power incision models, and the choice of characteristic scales. In my opinion, the discussion could also revisit some of the assumption behind the study (in particular uniform uplift).*

This would be an interesting discussion. Our characteristic scales depend on the parameters, so they would also become non-uniform if the parameters were non-uniform. Thus, we can add this discussion in Sect. 4.2, which deals with characteristic scales.

If the non-uniformity is gradual and follows a systematic pattern (e.g., the differential uplift across a fold described by Kirby and Whipple, 2001), then the resulting non-uniform characteristic scales could be useful. For example, designing a lab-scale sandbox landscape that models differential uplift might benefit from these non-uniform characteristic scales.

However, if the parameters were randomly heterogeneous, or they varied greatly over distances much smaller than typical landscape units, then the resulting "characteristic" scales might not be characteristic of anything, and thus they might lose their explanatory power.

*All in all, I think that the paper is already in a very good shape and ready to be published in ESURF after minor revisions.*

Thank you.

*Specific comments:*

*7-17 While referring the reader to Theodoratos et al. (2018), you may nevertheless provide some more details on the numeric simulations here, e.g. resolution or the nr of vertices used in the TIN.*

We will include some more details in an appendix.

*13-19 Given Eq. 1, this should rather read that points with any given stream power above the threshold value experience a stream power greater than zero, ...*

Thanks, we now see why this could be confusing. We will rephrase it.

**References**

Howard, A. D.: A detachment-limited model of drainage basin evolution, Water Resour. Res., 30(7), 2261– 2285, 1994.

Kirby, E. and Whipple, K.: Quantifying differential rock-uplift rates via stream profile analysis. Geology, 29(5), 415–418, 2001.

Mitášová, H. and Hofierka, J.: Interpolation by regularized spline with tension: II. Application to terrain modeling and surface geometry analysis, Math. Geol., 25(6), 657–669, https://doi.org/10.1007/BF00893172, 1993.

Theodoratos, N., Seybold, H. and Kirchner, J. W.: 2018. Scaling and similarity of a stream-power incision and linear diffusion landscape evolution model, Earth Surf. Dyn., 6(3), 779–808, https://doi.org/10.5194/esurf-6-779-2018, 2018.

Tucker, G. E.: Drainage basin sensitivity to tectonic and climatic forcing: Implications of a stochastic model for the role of entrainment and erosion thresholds, Earth Surf. Proc. Landforms, 29(2), 185–205, DOI: 10.1002/esp.1020, 2004.